# FastVID: Dynamic Density Pruning for Fast Video Large Language Models

**Leqi Shen**[1,2*]   **Guoqiang Gong**[3†]   **Tao He**[4]   **Yifeng Zhang**[3]   **Pengzhang Liu**[3]
**Sicheng Zhao**[2‡]   **Guiguang Ding**[1,2‡]
[1] School of Software, Tsinghua University   [2] BNRist, Tsinghua University
[3] JD.com   [4] GRG Banking Equipment Co., Ltd.

## Abstract

Video Large Language Models have demonstrated strong video understanding capabilities, yet their practical deployment is hindered by substantial inference costs caused by redundant video tokens. Existing pruning techniques fail to effectively exploit the spatiotemporal redundancy present in video data. To bridge this gap, we perform a systematic analysis of video redundancy from two perspectives: temporal context and visual context. Leveraging these insights, we propose Dynamic Density Pruning for Fast Video LLMs termed FastVID. Specifically, FastVID dynamically partitions videos into temporally ordered segments to preserve temporal structure and applies a density-based token pruning strategy to maintain essential spatial and temporal information. Our method significantly reduces computational overhead while maintaining temporal and visual integrity. Extensive evaluations show that FastVID achieves state-of-the-art performance across various short- and long-video benchmarks on leading Video LLMs, including LLaVA-OneVision, LLaVA-Video, Qwen2-VL, and Qwen2.5-VL. Notably, on LLaVA-OneVision-7B, FastVID effectively prunes **90.3%** of video tokens, reduces FLOPs to **8.3%**, and accelerates the LLM prefill stage by **7.1×**, while maintaining **98.0%** of the original accuracy. The code is available at `https://github.com/LunarShen/FastVID`.

## 1   Introduction

Video Large Language Models (Video LLMs) [7, 22, 19, 57, 45] have shown strong performance in video understanding but incur substantial inference costs. While several methods [38, 28, 48, 18, 37, 60] explore training-time compression to mitigate this issue, they often require expensive retraining. In this work, we focus on an inference-time acceleration strategy that enhances efficiency without requiring additional training.

This computational burden is primarily caused by the high volume of video tokens, making effective token compression essential. While prior image compression methods [6, 32, 56, 51, 5] reduce redundancy within a single image, they fail to exploit temporal dependencies across frames. As a result, spatiotemporal redundancy remain insufficiently explored. In this work, we systematically analyze video redundancy from two key perspectives: *temporal context* and *visual context*.

Temporal context is fundamental to video understanding, as the order and continuity of frames directly influence semantic interpretation. As depicted in Figure 1(a), disrupting frame order (shuffled) or omitting frames (incomplete) leads to incorrect comprehension, highlighting the necessity of preserving temporal structure. To achieve this, we segment the video into temporally ordered segments, grouping highly similar consecutive frames. Pruning is applied within each high-redundancy segment but not across segments, preserving the essential temporal structure for accurate video understanding.

---

*`lunarshen@gmail.com`, work done at JD.com.   †Project lead.   ‡ Corresponding authors.

39th Conference on Neural Information Processing Systems (NeurIPS 2025).

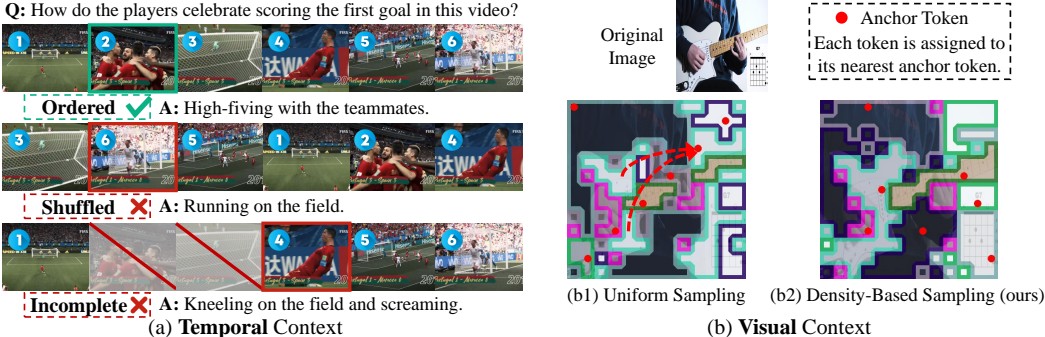

(a) **Temporal** Context     (b) **Visual** Context

Figure 1: Effective token compression in Video LLMs relies on preserving both temporal and visual context. (a) shows the impact of temporal disruptions, emphasizing the importance of temporal structure preservation. (b) show the effects of different token selection strategies for visual merging, where *patches that share the same inner and border color are merged*. Density-based Sampling retains both distinctive and representative context while effectively reducing redundancy.

Given high-redundancy segments, we aim to preserve visual context by effectively consolidating spatial and temporal information within each segment. A state-of-the-art method VisionZip [51] applies uniform token sampling followed by merging redundant tokens [5] as shown in Figure 1(b1). However, its token selection strategy is content-agnostic, potentially leading to the loss of important details, such as the guitar body being incorrectly merged with the background. To address this, we propose a density-based token sampling in Figure 1(b2). Specifically, high-density tokens, surrounded by numerous similar tokens, serve as candidates for selection. We select density peak tokens as anchors, which have a higher density than their neighbors and by a relatively large distance from tokens with higher densities [30]. This strategy ensures that the selected tokens are both representative and distinctive, effectively preserving segment visual context while reducing redundancy.

Building upon these insights, we propose Dynamic Density Pruning for **Fast VID**eo LLMs, minimizing spatiotemporal redundancy while preserving essential semantics. We begin with Dynamic Temporal Segmentation, which adaptively partitions videos into segments. Within each segment, we introduce Density Spatiotemporal Pruning to retain both global visual context and salient details. FastVID significantly accelerates inference while preserving both temporal and visual integrity.

To evaluate the generalization capability of our approach, we evaluate it on leading Video LLMs, LLaVA-OneVision [19], LLaVA-Video [57], Qwen2-VL [45], and Qwen2.5-VL [4]. To further validate its effectiveness, we perform extensive experiments on MVBench [21], LongVideoBench [46], MLVU [58], and VideoMME [12]. These benchmarks cover a wide range of video complexities and durations, ensuring a comprehensive evaluation. Notably, on LLaVA-OneVision-7B, FastVID effectively prunes **90.3%** of video tokens, reduces FLOPs to **8.3%**, and accelerates the LLM prefill stage by **7.1×**, while maintaining **98.0%** of the original accuracy across all benchmarks.

The main contributions are summarized as follows: (1) We analyze Video LLM compression from both temporal and visual context perspectives, emphasizing the importance of maintaining temporal and visual integrity. (2) We propose FastVID, a novel pruning framework that employs Dynamic Temporal Segmentation to partition videos into temporally ordered segments and Density Spatiotemporal Pruning to retain global segment information and key details. (3) Our FastVID achieves state-of-the-art performance across diverse Video LLMs and benchmarks, while maintaining robust accuracy even under extreme compression.

## 2 Related Work

**Video LLMs.** With the rapid advancement of LMMs [1, 8, 42] and MLLMs [43, 25, 26, 20, 3, 27, 15, 14], there has been growing interest in Video LLMs. These models can be categorized based on how they process video tokens: general Video LLMs and Video LLMs with training-time compression.

General Video LLMs [24, 7, 22, 19, 57, 45] directly process raw video tokens or apply pooling. Video-LLaVA [22] leverages shared projection layers to obtain unified visual representations. LLaVA-

OneVision [19] demonstrates strong video understanding through task transfer from images. LLaVA-Video [57] creates a high-quality synthetic dataset for video instruction-following. To better capture the spatiotemporal structure of video, some models [57, 45] introduce additional designs for video positional information. LLaVA-Video[1] introduces newline tokens to distinguish spatial and temporal positions effectively.

Video LLMs with training-time compression [38, 28, 48, 18, 37, 60] aim to significantly reduce the number of video tokens, enabling longer video sequences. Chat-UniVi [18] progressively clusters visual tokens and provides multi-scale features. LongVU [37] employs cross-modal query and inter-frame dependencies to adaptively reduce video redundancy. Apollo [60] explores scaling consistency and uses the Perceiver Resampler [17].

However, general Video LLMs remain the dominant paradigm, with LLaVA-OneVision and the Qwen-VL series being widely adopted due to its adaptability and superior performance. Therefore, we focus on inference-time acceleration for general Video LLMs.

**Token Compression.** Token compression has emerged as an effective approach to reduce computational complexity in transformer architectures, such as ViT [10] and CLIP [29]. ToMe [5] progressively merges fixed spatial tokens, while TempMe [35] extends this concept by merging neighboring clips to minimize temporal redundancy.

Recent studies [6, 32, 56, 51, 5, 47, 23, 59] primarily focus on spatial token reduction to accelerate Image LLMs. FastV [6] selects text-relevant tokens at shallow layers of the LLM. LLaVA-PruMerge [32] uses attention scores from the [CLS] token to prune spatial redundancy. SparseVLM [56] proposes a token recycling strategy to aggregate and reconstruct tokens. VisionZip [51] reduces visual redundancy in the vision encoders. However, these methods overlook the temporal relationships across frames.

Due to the high volume of video tokens in Video LLMs, recent video compression methods [13, 41, 16, 36, 34, 40] have gained increasing attention. FrameFusion [13] applies both merging and pruning across successive shallow LLM layers, but repeated pruning operations adversely affect overall efficiency. DyCoke [41] merges tokens across frames and applies dynamic KV cache reduction. However, its pruning during the prefill stage struggles to achieve substantial token reduction while maintaining accuracy. PruneVID [16] clusters video tokens and selects those most relevant to query tokens, but this dependency on clustering introduces significant latency during compression. Despite these advances, efficient and accurate pruning under large token reduction remains unsolved.

## 3 Methodology

### 3.1 FastVID

In this paper, we focus on preserving temporal and visual context at the prefill stage. By reducing video tokens before LLM processing, our FastVID significantly enhances computational efficiency while facilitating easy deployment, including compatibility with FlashAttention [9], KV cache, multi-turn conversations, and a plug-and-play design for seamless integration into existing Video LLMs. Our method achieves robust performance even under extreme compression rates, offering a practical solution for fast Video LLMs.

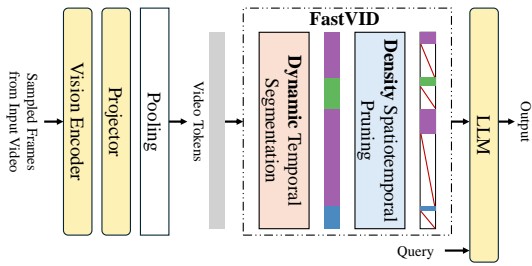

Figure 2: The overview of our FastVID.

Figure 2 presents an overview of FastVID. Given an input video, $F$ frames are uniformly sampled (*e.g.*, 32 in LLaVA-OneVision, 64 in LLaVA-Video). Each frame is individually processed by the vision encoder. The extracted tokens are projected and pooled into a video token sequence. With FastVID, we effectively eliminate redundant tokens while preserving critical information. Specifically, Dynamic Temporal Segmentation (DySeg) in Section 3.2 dynamically partitions video tokens into temporally ordered, high-redundancy segments, while Density Spatiotemporal Pruning (STPrune) in

---

[1]https://github.com/LLaVA-VL/LLaVA-NeXT

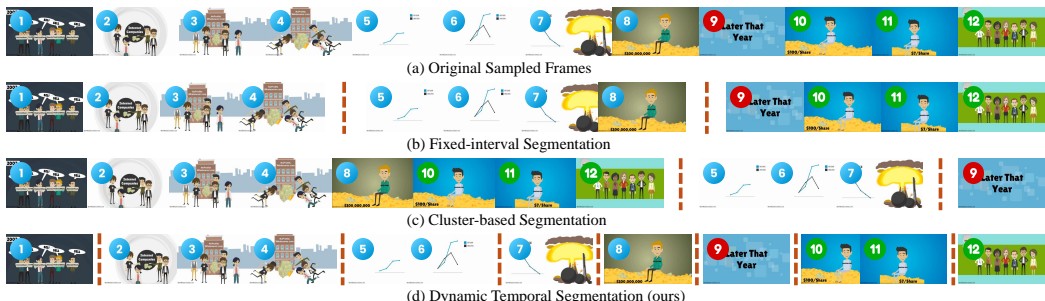

(a) Original Sampled Frames

(b) Fixed-interval Segmentation

(c) Cluster-based Segmentation

(d) Dynamic Temporal Segmentation (ours)

Figure 3: Visualization of different segmentation methods on 12 sampled frames from a video example. (b) Fixed-interval Segmentation struggles to maintain high intra-segment similarity, leading to visually diverse frames within the same segment. (c) Cluster-based Segmentation disrupts temporal order, grouping frames from different time periods to the same segment. (d) Our DySeg adaptively partitions the video, preserving both temporal structure and high intra-segment similarity. Please refer to Table 8 for a detailed quantitative comparison.

Section 3.3 performs density-based pruning within each segment, enhancing efficiency with minimal loss of key information. The final retained video tokens, combined with query tokens, is then fed into the LLM to generate the response.

## 3.2 Dynamic Temporal Segmentation

As shown in Figure 3(b) and 3(c), there are two common static segmentation methods for sampled frame sequences. In Figure 3(b), Fixed-interval Segmentation partitions the sequence into segments of a fixed length, preserving temporal order but potentially grouping visually dissimilar frames. Figure 3(c) shows Cluster-based Segmentation, where frames are grouped into three clusters based on frame similarity. However, it suffers from a predefined cluster number, leading to ineffective segmentation when video complexity varies. As a result, the first segment contains similar objects but different scenes. Furthermore, clustering may disrupt the temporal order by ignoring critical temporal information, such as omitting key frames (*e.g.*, the 9th frame) and incorrectly grouping frames from different time periods into the first segment.

To address the limitations of static methods, we propose Dynamic Temporal Segmentation, a simple yet effective method that adaptively refines segment boundaries according to video complexity. DySeg achieves both temporal structure and high intra-segment similarity, generating fewer partitions for simple scenes and finer ones for more complex scenes.

To enable effective pruning, DySeg induces high spatiotemporal redundancy within each segment by minimizing similarity between adjacent segments. Thus, we segment the video based on transition similarity between adjacent frames. Specifically, we utilize global frame features $\mathbf{f}$ to compute the cosine similarity:

$$t_i = \cos(\mathbf{f}_i, \mathbf{f}_{i+1}), \quad i = 1, \cdots, F-1,$$
$$\mathbf{T} = \{t_1, t_2, \cdots, t_{F-1}\}, \tag{1}$$

where $t_i$ denotes the transition similarity between the $i$-th and $(i+1)$-th frame. $\mathbf{T}$ denotes the $F-1$ transition similarities for $F$ sampled frames. To achieve dynamic segmentation, we select transitions that satisfy the following conditions:

$$\mathbf{S}_1 = \arg\min_{c-1} \mathbf{T}, \quad \mathbf{S}_2 = \{i \mid t_i < \tau, \ t_i \in \mathbf{T}\},$$
$$\mathbf{S} = \mathbf{S}_1 \cup \mathbf{S}_2, \tag{2}$$

where $c$ denotes the minimum number of segments and $\tau$ denotes the threshold for transition similarity. $\mathbf{S}_1$ denotes the $c-1$ most dissimilar frame transitions, while $\mathbf{S}_2$ denotes transitions where similarity falls below a fixed threshold $\tau$. Each transition in the union $\mathbf{S}$ marks a boundary between segments. In simple videos with minimal shot transitions, $\mathbf{S}_2$ is often empty. In such cases, $\mathbf{S}_1$ enables finer segmentation by distinguishing subtle temporal changes. For complex videos, $\mathbf{S}_1$ is typically a subset of $\mathbf{S}_2$, where $\mathbf{S}_2$ ensures that adjacent frames with similarity below $\tau$ are assigned to different segments. In Figure 3(d), our DySeg effectively segments videos by dynamically adjusting granularity in a simple yet effective manner, outperforming Fixed-interval and Clustering-based methods.

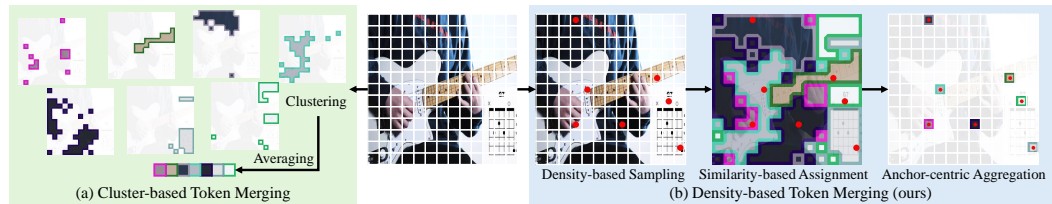

Figure 4: Comparison of our proposed DTM and Cluster-based Token Merging. (a) Cluster-based methods aggregate tokens within each cluster and concatenate them, leading to a loss of positional information and disruption of the spatiotemporal structure in video data. (b) Our DTM updates anchor tokens while preserving their original positional information, maintaining structural coherence. Please refer to Table 7 for a detailed quantitative comparison.

### 3.3 Density Spatiotemporal Pruning

After obtaining segments with highly similar frames, we introduce STPrune to reduce redundant tokens. It consists of two key modules: Density-based Token Merging (DTM) for segment visual context and Attention-based Token Selection (ATS) for salient details. For a segment of $P$ frames, we retain $rPN$ tokens in total, where $N$ is the number of tokens per frame, and $r$ denotes the retention ratio. These tokens are generated by DTM and ATS. Specifically, $drPN$ tokens are generated by DTM and $(1-d)rPN$ tokens are generated by ATS, where $d$ controls the token allocation between the two modules.

**Density-based Token Merging.** For segment visual context, we first identify anchor tokens and to-be-merged tokens. Selecting anchor tokens from the entire segment would disrupt their spatial relationships, as anchor tokens corresponding to different components of the same object might be distributed across multiple frames. To mitigate this, we restrict anchor token selection to specific frames. Specifically, we sample anchor frames at a fixed interval $p$ and select anchor tokens from these frames. The remaining tokens in the segment are treated as to-be-merged tokens. Specifically, the number of anchor frames is $\lceil P/p \rceil$. From each anchor frame, $drPN/\lceil P/p \rceil$ tokens are selected as anchor tokens. Please see Figure 5 for a visualization of DTM applied to video segments.

To further clarify DTM, Figure 4(b) illustrates DTM when the segment length is 1 and shows that it consists of three key steps. First, we follow density peaks clustering algorithms [30, 11] to compute the density score. For each token in the anchor frame $[v_1, v_2, \cdots, v_N]$, we calculate its local density $\rho_i$ and its distance to the closest higher-density token $\delta_i$, obtaining the final density score $\rho_i \times \delta_i$:

$$\rho_i = \exp(-\frac{1}{k} \sum_{v_j \in \text{kNN}(v_i)} \text{d}(v_i, v_j)^2), \tag{3}$$

$$\delta_i = \begin{cases} \min_{j:\rho_j > \rho_i} \text{d}(v_i, v_j), & \text{if } \exists j \text{ s.t. } \rho_j > \rho_i \\ \max_{j} \text{d}(v_i, v_j), & \text{otherwise} \end{cases} \tag{4}$$

where $\text{d}(v_i, v_j)$ denotes the Euclidean distance. Density peak tokens with high $\rho_i \times \delta_i$ serve as anchor tokens, indicating that they are surrounded by neighbors with lower local density while remaining relatively distant from other high-density tokens. This selection ensures that anchor tokens are both representative and distinctive. Next, for each anchor frame, Similarity-based Assignment assigns each token in the segment to the nearest anchor token using cosine similarity. Finally, we apply Anchor-centric Aggregation to merge the assigned tokens into their respective anchors, preserving key visual structures through representative tokens. For an anchor $a$ and its associated tokens $[b_1, \cdots, b_n]$, the updated $a^*$ is computed as:

$$a^* = \beta a + \frac{1-\beta}{n} \sum_{i=1}^{n} b_i, \tag{5}$$

where $\beta$ controls the balance between the anchor token and its associated tokens. In Figure 4, we compare our DTM with Cluster-based Token Merging. In LLMs, RoPE [39] encodes relative positional relationships between tokens, making positional information essential for maintaining the spatiotemporal structure of video tokens. While prior methods [16] rely on Cluster-based Token

Table 1: Comparison of state-of-the-art methods on LLaVA-OneVision [19]. The A%/B% retention ratio indicates that A% of the LLM input tokens are retained, and subsequently compressed to B% during the LLM forward pass. The best performance among those with similar retention ratios $R$ is highlighted in bold. TFLOPs related to video tokens are reported (see Appendix A for details).

| Method | Retention Ratio $R$ | TFLOPs | MVBench | LongVideo Bench | MLVU | VideoMME | | | | Avg. Acc. | |
|---|---|---|---|---|---|---|---|---|---|---|---|
| | | | | | | Overall | Short | Medium | Long | Score | % |
| Duration | | | 16s | 1~60min | 3~120min | 1~60min | 1~3min | 3~30min | 30~60min | | |
| Vanilla | 100% | 48.82 | 56.9 | 56.4 | 65.2 | 58.6 | 70.3 | 56.6 | 48.8 | 59.3 | 100 |
| DyCoke [41]ᴄᴠᴘʀ'25 | 32.5% | 14.13 | 56.3 | 56.6 | 62.1 | 57.1 | 68.1 | 56.7 | 46.7 | 58.0 | 97.8 |
| FastV [6]ᴇᴄᴄᴠ'24 | 100%/25% | 13.45 | 54.7 | 55.5 | 61.5 | 56.2 | 68.0 | 54.6 | 46.0 | 57.0 | 96.1 |
| VisionZip [51]ᴄᴠᴘʀ'25 | 25% | 10.73 | 53.7 | 51.2 | 58.5 | 54.1 | 61.6 | 53.4 | 47.2 | 54.4 | 91.7 |
| VisionZip* [51]ᴄᴠᴘʀ'25 | 25% | 10.73 | **56.6** | 55.7 | **64.8** | **58.0** | 68.6 | **57.7** | 47.7 | **58.8** | **99.1** |
| DyCoke [41]ᴄᴠᴘʀ'25 | 25% | 10.73 | 49.5 | 48.1 | 55.8 | 51.0 | 61.1 | 48.6 | 43.2 | 51.1 | 86.2 |
| **FastVID** | 25% | 10.73 | 56.5 | **56.3** | 64.1 | **58.0** | **69.9** | 56.6 | **47.7** | 58.7 | 99.0 |
| FastV [6]ᴇᴄᴄᴠ'24 | 100%/20% | 11.38 | 54.1 | 56.6 | 61.2 | 56.2 | 66.8 | 54.6 | 47.2 | 57.0 | 96.1 |
| VisionZip [51]ᴄᴠᴘʀ'25 | 19.9% | 8.46 | 53.0 | 50.0 | 57.1 | 53.0 | 60.8 | 51.0 | 47.1 | 53.3 | 90.0 |
| VisionZip* [51]ᴄᴠᴘʀ'25 | 19.9% | 8.46 | 55.8 | 55.4 | **64.2** | **58.0** | 68.6 | **57.0** | 48.3 | 58.4 | 98.5 |
| **FastVID** | 19.9% | 8.46 | **56.3** | **57.1** | 63.9 | 57.9 | **69.3** | 56.7 | 47.7 | **58.8** | **99.1** |
| FastV [6]ᴇᴄᴄᴠ'24 | 100%/15% | 9.35 | 53.2 | 54.9 | 59.8 | 54.7 | 65.1 | 53.4 | 45.7 | 55.7 | 93.9 |
| VisionZip [51]ᴄᴠᴘʀ'25 | 14.8% | 6.23 | 50.3 | 46.9 | 54.4 | 49.5 | 55.8 | 49.3 | 43.3 | 50.3 | 84.8 |
| VisionZip* [51]ᴄᴠᴘʀ'25 | 14.8% | 6.23 | 54.3 | 53.9 | 63.1 | 55.5 | 63.0 | 54.4 | **49.1** | 56.7 | 95.6 |
| **FastVID** | 14.8% | 6.23 | **56.0** | **56.2** | **63.2** | **57.7** | **69.3** | **56.2** | 47.4 | **58.3** | **98.3** |
| FastV [6]ᴇᴄᴄᴠ'24 | 100%/10% | 7.36 | 51.7 | 52.1 | 57.7 | 52.4 | 60.9 | 51.4 | 45.0 | 53.5 | 90.2 |
| VisionZip [51]ᴄᴠᴘʀ'25 | 9.7% | 4.04 | 44.4 | 43.5 | 51.5 | 46.0 | 50.4 | 45.8 | 41.8 | 46.4 | 78.3 |
| VisionZip* [51]ᴄᴠᴘʀ'25 | 9.7% | 4.04 | 51.7 | 48.3 | 59.7 | 52.8 | 59.4 | 52.0 | 46.9 | 53.1 | 89.6 |
| PruneVID* [16]ᴀᴄʟ'25 | 10.1% | 4.23 | 54.2 | 53.8 | 62.3 | 55.9 | 66.4 | 52.9 | 48.3 | 56.6 | 95.4 |
| **FastVID** | 9.7% | 4.04 | **55.9** | **56.3** | **62.7** | **57.3** | **67.4** | **56.0** | **48.6** | **58.1** | **98.0** |
| PruneVID [16]ᴀᴄʟ'25 | 10.1%/3.9% | 2.58 | 54.1 | 51.8 | **62.3** | 55.5 | **67.1** | 51.8 | 47.7 | 55.9 | 94.3 |
| **FastVID**+FastV [6] | 9.7%/3.6% | 2.47 | **55.0** | **53.6** | 61.9 | **56.3** | 66.1 | **54.8** | **48.1** | **56.7** | **95.6** |

Merging, they discard positional information, causing RoPE to struggle with encoding spatiotemporal structure. In contrast, our DTM maintains the positional information of merged tokens, enhancing visual context understanding.

Overall, our DTM offers three key advantages: (1) Density-based Sampling selects density peak tokens as anchors, ensuring representative visual context. (2) We maintain the original positional information of updated anchor tokens, preserving spatiotemporal structure. (3) Anchor-centric Aggregation emphasizes representative tokens, enhancing feature representation.

**Attention-based Token Selection.** In addition to segment visual context obtained through DTM, we introduce ATS to capture salient visual details.

Motivated by previous studies [32, 51, 44, 55], we utilize [CLS] attention scores to identify salient visual information. However, in Video LLMs, which commonly use SigLIP [53] as their vision encoder, [CLS] attention scores cannot be obtained. This is because Video LLMs utilize the penultimate layer of SigLIP, omitting the SigLIP head where the [CLS] token is generated. To overcome this, we reintegrate a pretrained SigLIP head into the vision encoder, allowing the model to compute [CLS] attention scores. Since the SigLIP head is lightweight (15.2M parameters), this modification incurs minimal computational overhead compared to the full Video LLM.

Specifically, we extract the [CLS] attention score from the pretrained SigLIP head, $\mathbf{A} \in \mathbb{R}^{H \times W}$, where $H$ and $W$ are the spatial dimensions of frame tokens. Since Video LLMs incorporate pooling (see Figure 2), we apply the same operation to $\mathbf{A}$, resulting in $\bar{\mathbf{A}} \in \mathbb{R}^{\bar{H} \times \bar{W}}$, where $\bar{H}$ and $\bar{W}$ are the pooled spatial dimensions. Finally, for each frame, we select the top $(1 - d)rN$ tokens with the highest attention scores to preserve critical visual details.

## 4 Experiments

### 4.1 Experimental Settings

**Benchmarks.** We evaluate our method on several widely used video understanding benchmarks: MVBench [21, 31], LongVideoBench [46], MLVU [58], and VideoMME (wo sub.) [12]. Specifically, VideoMME is officially divided into short, medium, and long subsets. These benchmarks contain videos of varying durations and complex scenarios, providing a comprehensive evaluation of our method's effectiveness and generalization.

**Implementation Details.** We apply our method to leading Video LLMs: LLaVA-OneVision [19], LLaVA-Video [57], Qwen2-VL [45], and Qwen2.5-VL [4]. Unless otherwise specified, we adopt

Table 2: Comparison of state-of-the-art methods on LLaVA-Video [57]. TFLOPs* calculations include both video and newline tokens.

| Method | Retention Ratio $R$ | # Newline Tokens $M$ | TFLOPs* | MVBench | LongVideo Bench | MLVU | VideoMME | | | Avg. Acc. | |
| --- | --- | --- | --- | --- | --- | --- | --- | --- | --- | --- | --- |
| | | | | | | | Overall | Short | Long | Score | % |
| Vanilla | 100% | 832 | 103.2 | 60.4 | 59.6 | 70.3 | 64.1 | 76.9 | 53.4 | 63.6 | 100 |
| DyCoke [41]CVPR'25 | 32.1% | 256 | 27.1 | 59.3 | 57.9 | 65.7 | 61.6 | 74.6 | 51.1 | 61.1 | 96.1 |
| FastV [6]ECCV'24 | 100%/25% | 832/568.7 | 29.2 | 58.0 | **58.3** | 63.9 | 61.0 | 71.3 | 51.0 | 60.3 | 94.8 |
| VisionZip [51]CVPR'25 | 24.9% | 64 | 19.5 | 56.4 | 54.1 | 62.1 | 58.6 | 66.3 | 51.2 | 57.8 | 90.9 |
| VisionZip* [51]CVPR'25 | 24.9% | 64 | 19.5 | 58.3 | **58.3** | 66.6 | 61.9 | 73.3 | 52.2 | 61.5 | 96.7 |
| DyCoke [41]CVPR'25 | 25% | 208 | 20.7 | 50.8 | 53.0 | 56.9 | 56.1 | 65.8 | 48.9 | 54.2 | 85.2 |
| **FastVID*** | 24.9% | 64 | 19.5 | 59.3 | **58.3** | 67.7 | 62.6 | **74.9** | 52.0 | 62.0 | 97.5 |
| **FastVID** | 24.9% | 715.5 | 24.5 | **59.9** | 57.4 | **68.6** | 63.6 | **74.9** | 53.7 | 62.4 | 98.1 |
| FastV [6]ECCV'24 | 100%/10% | 832/311.8 | 16.2 | 55.8 | 55.4 | 58.9 | 57.9 | 67.6 | 48.6 | 57.0 | 89.6 |
| VisionZip [51]CVPR'25 | 9.5% | 64 | 7.3 | 46.3 | 46.6 | 52.2 | 49.5 | 54.2 | 44.3 | 48.7 | 76.6 |
| VisionZip* [51]CVPR'25 | 9.5% | 64 | 7.3 | 56.6 | 53.6 | 61.7 | 58.7 | 67.6 | 50.1 | 57.7 | 90.7 |
| **FastVID*** | 9.5% | 64 | 7.3 | 58.3 | 56.2 | 63.9 | 59.6 | 70.9 | 50.7 | 59.5 | 93.6 |
| **FastVID** | 9.5% | 508.2 | 10.5 | **58.5** | **56.5** | **64.9** | **60.7** | **71.7** | **51.2** | **60.2** | **94.7** |

Table 3: Comparison of state-of-the-art methods on Qwen2-VL [45]. We set the maximum number of video tokens fed into the LLM to 16384, and the maximum number of sampled frames to 768. FastV performs pruning based on LLM attention scores, but its eager-attention implementation materializes the full attention matrix in memory, leading to OOM errors due to the large number of video tokens.

| Method | # Token | | TFLOPs | | VideoMME | | | | |
| --- | --- | --- | --- | --- | --- | --- | --- | --- | --- |
| | | | | | Short | Medium | Long | Overall | |
| Vanilla | 13447.1 | 100% | 124.0 | 100% | 74.1 | 60.4 | 54.3 | 63.0 | 100% |
| FastV [6]ECCV'24 | 13447.1/3361.8 | 100%/25% | 31.3 | 25.3% | Out of Memory | | | | |
| VisionZip* [51]CVPR'25 | 3349.3 | 24.9% | 24.1 | 19.4% | 70.9 | 56.3 | 48.3 | 58.5 | 92.9% |
| PruneVID* [16]ACL'25 | 3460.6 | 25.7% | 25.0 | 20.1% | 66.7 | 54.0 | 48.1 | 56.3 | 89.4% |
| **FastVID** | 3349.3 | 24.9% | 24.1 | 19.4% | **72.7** | **58.3** | **50.7** | **60.6** | **96.2%** |

the hyperparameter setting $c = 8, \tau = 0.9, d = 0.4, p = 4, \beta = 0.6$ for all experiments. For LLaVA-OneVision, 32 sampled frames generate a $32 \times 196$ token input to the LLM. We experiment with $r \in \{25\%, 20\%, 15\%, 10\%\}$. For LLaVA-Video, 64 sampled frames generate a $64 \times 169$ token input. We experiment with $r \in \{25\%, 10\%\}$. For the Qwen-VL series, which samples up to 768 frames, we discard highly redundant frames and set $\tau$ to its optimal value. We conduct all evaluations using LMMs-Eval [54] on A100 GPUs.

**Compared Baselines.** (1) For image compression, we adopt both the widely used classic method FastV [6] and the current state-of-the-art method VisionZip [51]. VisionZip prunes tokens in the last layer of the MLLM's ViT, conflicting with pooling in Video LLMs and degrading performance. To address this, we implement VisionZip*, which applies pruning after pooling. (2) For video compression, we compare two recent methods, DyCoke [41] and PruneVID [16]. Our FastVID focuses on pruning during the prefill stage. To ensure fairness, we reimplement these baselines without pruning in the decode stage. PruneVID applies two-stage pruning on both input tokens and the LLM's 10th layer. PruneVID* refers to the variant that prunes only input tokens. More implementation details of these baselines are provided in Appendix A.

## 4.2 Comparisons with State-of-the-Art Methods

For a comprehensive evaluation, we compare our FastVID with state-of-the-art methods on benchmarks with diverse video durations. We perform evaluations across different retention ratios $R$ to systematically assess model performance. In the case of DyCoke, video frames are evenly divided into 4-frame segments, retaining all tokens from the first frame while pruning the rest. Consequently, its lowest retention ratio $R$ is 25%.

**Results on LLaVA-OneVision.** In Table 1, we evaluate our FastVID against other methods on LLaVA-OneVision. While FastV and VisionZip perform well at $R = 25\%$, but their performance declines sharply as $R$ decreases. This indicates the limitations of spatial compression alone, which struggles to preserve essential temporal information under extreme pruning. Notably, even after pruning **90.3%** of the tokens, our FastVID preserves **98.0%** of the vanilla model's performance. To compare with PrunVID, we additionally apply FastV at the 10th layer of the LLM. FastVID+FastV achieves **95.6%** of the original accuracy at **2.47** (**5.1%**) TFLOPs.

Table 4: Comparison of state-of-the-art methods on Qwen2.5-VL [4]. We set the maximum number of video tokens fed into the LLM to 16384, and the maximum number of sampled frames to 768.

| Method | # Token | | TFLOPs | | VideoMME | | | | |
|---|---|---|---|---|---|---|---|---|---|
| | | | | | Short | Medium | Long | Overall | |
| Vanilla | 13447.1 | 100% | 124.0 | 100% | 76.7 | 68.2 | 56.9 | 67.3 | 100% |
| PruneVID* [16]$_{ACL'25}$ | 3294.5 | 24.5% | 23.7 | 19.1% | 67.0 | 59.4 | 51.6 | 59.3 | 88.1% |
| **FastVID** | 3240.7 | 24.1% | 23.2 | 18.7% | **74.3** | **61.3** | **52.8** | **62.8** | **93.3%** |

Table 5: Efficiency Comparison on LLaVA-OneVision [19]. The prefill time, defined as the latency to the first generated token, is measured on VideoMME using an A100 GPU.

| Method | # Token | TFLOPs | Prefill Time (ms) | | | | Avg. Acc. |
|---|---|---|---|---|---|---|---|
| | | | Segmentation | Compression | LLM Forward | Total | |
| Vanilla | 6272 (100%) | 48.82 (100%) | – | – | 476.3 | 476.3 (1.0×) | 59.3 (100%) |
| PruneVID* [16] | 635.6 (10.1%) | 4.23 (8.7%) | 5.2 | 32.0 | 64.3 | 101.5 (4.7×) | 56.6 (95.4%) |
| FastVID | **608 (9.7%)** | **4.04 (8.3%)** | **0.5** | **5.6** | **61.1** | **67.2 (7.1×)** | **58.1 (98.0%)** |

**Results on LLaVA-Video.** LLaVA-Video adopts a unique design by inserting newline tokens after each height-wise position in every frame, producing $64 \times 13 \times (13 + 1)$ tokens. In Table 2, for all methods, when the positional information of retained tokens is preserved, we also retain the associated newline tokens. For a fair comparison with VisionZip, we evaluate FastVID*, where only one newline token is retained per frame. Notably, our FastVID consistently outperforms all baselines across various retention ratios.

**Results on Qwen2-VL.** Unlike LLaVA-OneVision [19] and LLaVA-Video [57], Qwen2-VL [45] employs a distinct architecture that samples 768 frames per video and processes them using 3D convolutions. It introduces M-RoPE, which decomposes rotary embeddings into temporal, height, and width components. The model dynamically adjusts the resolution of each frame. In Table 3, our FastVID outperforms other baselines on Qwen2-VL under similar model complexity. In particular, FastVID achieves a **2.4** accuracy improvement on the long subset.

**Results on Qwen2.5-VL.** To further demonstrate its generalization ability, we report results on Qwen2.5-VL [4] in Table 4. Unlike the LLaVA series and Qwen2-VL, Qwen2.5-VL employs a vision encoder with window attention and a Qwen2.5 LLM [50], representing a substantially different architecture. Compared to the recent SOTA PruneVID, FastVID achieves a **3.5** improvement under a comparable retention ratio. As shown in Tables 1, 2, 3, and 4, our FastVID's consistent superiority across these distinct Video LLM architectures further validates its strong generalizability and practical effectiveness.

**Efficiency Comparison.** Table 5 compares the efficiency of our method against the state-of-the-art video compression method PruneVID. Latency and accuracy are evaluated while maintaining similar model complexity. PruneVID relies heavily on time-consuming clustering algorithms during both video segmentation and compression. In contrast, FastVID leverages efficient transition similarity to achieve efficient segmentation. Although density score computation (see Eq. (3-4)) is time-consuming, we restrict this step to anchor frames and parallelize its execution, thereby accelerating the compression process. As a result, FastVID achieves a **7.1**× speedup while preserving **98.0%** accuracy. Further comparisons with PruneVID are provided in Appendix C.

### 4.3 Ablation Study

By default, we conduct ablation studies on LLaVA-OneVision at $r = 10\%$. Further hyperparameter analysis is provided in Appendix B.

**Ablation study on DySeg.** To compare different video segmentation methods, we present qualitative and quantitative results in Figure 3 and Table 8, respectively. In Table 8, Fixed-interval Segmentation with an interval of $4$ generates $32/4 = 8$ segments, while Cluster-based Segmentation generates 8 clusters, ensuring consis-

Table 8: Ablation study on video segmentation.

| Segmentation | MVBench | LongVideo-Bench | MLVU | VideoMME | Avg. Acc. |
|---|---|---|---|---|---|
| Fixed-interval | 55.1 | 53.6 | 61.7 | 55.4 | 95.3% |
| Cluster-based | 53.2 | 53.0 | 61.7 | 54.8 | 93.9% |
| Our DySeg | **55.9** | **56.3** | **62.7** | **57.3** | **98.0%** |

Table 6: Ablation Study on $d$ in STPrune. STPrune consists of DTM and ATS. A fraction $d$ of the retained tokens is from DTM, while the remaining $1 - d$ is from ATS.

| $d$ in STPrune | MVBench | LongVideo-Bench | MLVU | VideoMME | Avg. Acc. |
|---|---|---|---|---|---|
| 0.0/ATS | 55.3 | 55.3 | 62.4 | 55.0 | 96.2% |
| 0.2 | 55.0 | 53.6 | 62.1 | 56.1 | 95.7% |
| 0.4 | 55.9 | **56.3** | **62.7** | **57.3** | **98.0%** |
| 0.6 | 55.6 | 54.5 | 62.0 | 56.2 | 96.3% |
| 0.8 | **56.0** | 54.9 | 62.6 | 55.6 | 96.7% |
| 1.0/DTM | 54.1 | 50.1 | 61.5 | 55.9 | 93.5% |

Table 7: Ablation study on different token merging strategies. Figure 1(b1) presents Uniform. Figure 4(a) presents Cluster-based.

| Token Merging | MVBench | LongVideo-Bench | MLVU | VideoMME | Avg. Acc. |
|---|---|---|---|---|---|
| LLaVA-OneVision | | | | | |
| Uniform | 55.0 | 54.7 | 62.0 | 56.4 | 96.2% |
| Cluster-based | 55.6 | 55.2 | 62.4 | **57.3** | 97.2% |
| Our DTM | **55.9** | **56.3** | **62.7** | 57.3 | **98.0%** |
| LLaVA-Video | | | | | |
| Uniform | 55.4 | 54.6 | 62.9 | 59.8 | 91.5% |
| Cluster-based | 55.3 | 55.1 | 62.9 | 60.1 | 91.8% |
| Our DTM | **58.5** | **56.5** | **64.9** | **60.7** | **94.7%** |

Table 9: Ablation study on the SigLIP head in ATS.

| Method | VideoMME |
|---|---|
| FastVID | **57.3** |
| w/o the SigLIP head | 56.3 |

Table 10: Improved length extrapolation via FastVID.

| Method | # Frame | # Token | VideoMME | |
|---|---|---|---|---|
| Vanilla | 32 | 6272 | 58.6 | 100% |
| FastVID (r=25%) | 128 | 6272 | 60.4 | 103.1% |
| FastVID (r=10%) | 320 | 6080 | 61.4 | 104.8% |

tency with the minimal segment number $c$ in our DySeg. Fixed-interval preserves temporal order between segments, whereas Cluster-based maintains intra-segment similarity. The results show that Fixed-interval outperforms Cluster-based by $1.4\%$, highlighting the importance of temporal structure preservation in Video LLM pruning. Additionally, high intra-segment similarity is also essential for effective pruning. Our proposed DySeg successfully integrates both advantages while enabling dynamic pruning, achieving a superior average performance of **98.0%**.

**Ablation study on STPrune.** STPrune is proposed to prune tokens within each segment and consists of two key components: DTM and ATS. DTM employs density-based sampling to merge redundant tokens, preserving segment visual context. ATS leverages [CLS] attention scores to highlight important visual details. Table 6 presents an ablation study on $d$, which controls the token distribution between DTM and ATS. The results show that ATS alone (when $d = 0.0$) outperforms DTM alone (when $d = 1.0$). However, the best performance is achieved at $d = 0.4$, demonstrating that a balanced integration of both modules is crucial for preserving essential video information. This integration boosts performance by **1.8%** and **4.5%** points over ATS and DTM alone, respectively.

**Ablation study on DTM.** In Table 7, we compare different token merging strategies for segment visual context. Uniform, used in VisionZip, selects anchor tokens in a content-agnostic manner, which limits its ability to distinguish semantically meaningful objects. Cluster-based discards essential positional information, resulting in inferior performance, particularly in LLaVA-Video. By leveraging Density-based Sampling and Anchor-centric Aggregation, our DTM achieves superior results, especially in LLaVA-Video, where DTM outperforms other methods by **2.9%**.

**Ablation study on the SigLIP head in ATS.** We reintroduce the original, un-finetuned SigLIP head to generate attention scores in ATS. This creates a potential parameter gap between the fine-tuned SigLIP encoder and the original SigLIP head. However, we believe this gap has limited impact for the following reasons.

First, the SigLIP head is pretrained jointly with the encoder on large-scale vision-language data, enabling strong generalization and transferability. Second, since the first-stage training of Video LLMs also aims to align visual and textual representations, the SigLIP head is naturally compatible with this alignment objective. Finally, as shown in Table 9, results show that using the SigLIP head leads to a 1.0 improvement, indicating its effectiveness. This is likely because the SigLIP head is explicitly trained to aggregate patch tokens for semantic alignment, making its attention scores better suited for identifying semantic salient tokens.

In addition, we emphasize that our FastVID generalizes to vision encoders without pretrained heads or [CLS] tokens (e.g., the Qwen-VL series). In such cases, [CLS] attention scores are computed at the Video LLM's ViT final layer using pseudo [CLS] tokens derived by averaging patch tokens. Our FastVID achieves strong performance on Qwen2-VL (Table 3) and Qwen2.5-VL (Table 4), significantly outperforming previous methods.

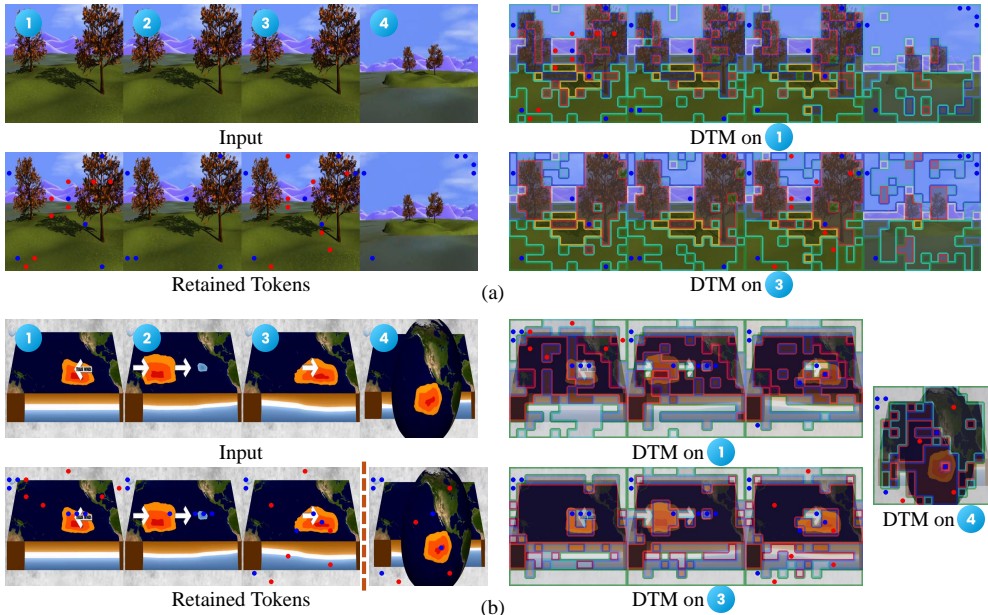

Figure 5: Visualization of FastVID with $c = 0, \tau = 0.9, d = 0.4, p = 2$. We retain a total of 40 tokens, including 16 tokens in DTM (highlighted in red) and 24 in ATS (highlighted in blue). In DTM, patches that share the same inner and border color are merged.

**Improved length extrapolation.** In Table 10, by applying FastVID with different retention ratios, we extend the number of sampled frames from 32 to 320 while maintaining a comparable total number of input video tokens. This consistently improves performance, which demonstrates that FastVID not only compresses video tokens efficiently but also enables effective length extrapolation.

### 4.4 Qualitative Results

Figure 5 visualizes the proposed DySeg and STPrune. In Figure 5(a), all frames are grouped into a single segment, with the 1st and 3rd frames selected as anchor frames, where DTM is applied. In Figure 5(b), the frames are divided into two segments, with the 1st, 3rd, and 4th frames selected as anchor frames. Blue tokens represent those selected by ATS due to high [CLS] attention. These tokens readily cluster together and do not always correspond to object regions, suggesting that while they reflect salient [CLS] information, they lack broader visual context. In contrast, DTM's red tokens effectively aggregate visually similar content across frames, significantly reducing spatiotemporal redundancy while preserving visual context. Together, ATS and DTM offer complementary benefits for effective video compression. More qualitative results are provided in Appendix B.

## 5 Conclusion

In this paper, we introduced FastVID, a novel inference-time pruning framework designed to accelerate Video LLMs by effectively reducing spatiotemporal redundancy. Through a comprehensive analysis of video tokens from both temporal context and visual context, FastVID dynamically partitions videos into temporally ordered segments and employs density-based token pruning within each segment. Extensive experiments across multiple Video LLMs and benchmarks demonstrate its generalization ability and effectiveness. Crucially, FastVID maintains high performance even under extreme compression rates, enabling practical deployment of fast Video LLMs.

## Acknowledgments

This work was supported by National Natural Science Foundation of China (Nos. 62525103, 62571294), Beijing Natural Science Foundation (No. L252009), CCF-DiDi GAIA Collaborative Research Funds, and the Postdoctoral Science Foundation of China (No. 2024M750565).

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

# Appendix

This appendix provides additional details and results to support our main paper:

- Appendix A presents additional experimental settings, including reproduction details of the compared baselines and an estimation of the computational cost.

- Appendix B presents additional experimental results, such as ablation studies on key hyperparameters and further quantitative and qualitative results.

- Appendix C discusses limitations of our approach and its potential broader impacts.

## A  Additional Experimental Settings

### A.1  Reproduction Details of Compared Baselines

All experiments are conducted using LMMs-Eval[2] [54] for consistency. The performance of the vanilla versions of LLaVA-OneVision[3] [19], LLaVA-Video[3] [57], and Qwen2-VL[4] [45] differs slightly from their reported results, remaining within an acceptable margin of error. We reimplement all baseline methods using LMMs-Eval, following their official implementations:

- **FastV**[5] [6] (**ECCV 2024**). FastV performs token pruning at the $K$-th layer of the LLM using attention scores, with a filtering ratio $R$. We reimplement it with $K = 2$, using $R \in \{75\%, 80\%, 85\%, 90\%\}$ in Table 1 and $R \in \{75\%, 90\%\}$ in Table 2.

- **VisionZip**[6] [51] (**CVPR 2025**). Visionzip performs pruning at the vision encoder's output, which conflicts with pooling operations in Video LLMs and degrades performance. To address this, we implement VisionZip*, which applies pruning after pooling. Following the original settings, each frame retains both dominant and contextual tokens in a $54\!:\!10$ ratio. We define $r$ as the proportion of tokens retained per frame. We set $r \in \{25\%, 20\%, 15\%, 10\%\}$ in Table 1 and $r \in \{25\%, 10\%\}$ in Table 2.

- **DyCoke**[7] [41] (**CVPR 2025**). DyCoke prunes during both the prefill and decode stages. For fair comparison, we only evaluate its prefill stage. The pruning rate in the TTM module is set $K \in \{0.9, 1.0\}$ in both Table 1 and Table 2.

- **PruneVID**[8] [16] (**ACL 2025**). PruneVID includes both input-stage and intra-LLM pruning in the prefill phase, along with decoding-stage pruning. For fair comparison, we evaluate two variants: PruneVID* (input-stage pruning only) and PruneVID (full prefill-stage pruning). Following the original settings, we use a threshold $\tau = 0.8$, temporal segment ratio $\gamma = 0.25$, token selection ratio $\alpha = 0.4$, and attention calculations use the 10th layer. The cluster ratio $\beta$ controls compression in the input-stage pruning. We use $\beta = 11\%$ in Table 1 and Table 5.

### A.2  Computational Cost Estimation

Following prior works [6, 47], we report the theoretical FLOPs of the LLM related to visual/video tokens. Specifically, LLaVA-OneVision [19], LLaVA-Video [57], and Qwen2-VL [45] are all built on Qwen2 [49], which employs grouped-query attention [2] and a SwiGLU-based three-layer feed-forward network [33]. The per-layer FLOPs of the LLM are computed as:

$$2nD(h_{kv}d) + 2nD^2 + 2n^2D + 3nDD' \tag{6}$$

where $n$ is the number of video tokens, $D$ is the hidden state size, $D'$ is the FFN intermediate size, $h_{kv}$ is the number of key/value heads, and $d$ is the head dimension.

---

[2] https://github.com/EvolvingLMMs-Lab/lmms-eval, MIT License

[3] https://github.com/LLaVA-VL/LLaVA-NeXT, Apache License 2.0

[4] https://github.com/QwenLM/Qwen2.5-VL, Apache License 2.0

[5] https://github.com/pkunlp-icler/FastV

[6] https://github.com/dvlab-research/VisionZip, Apache License 2.0

[7] https://github.com/KD-TAO/DyCoke, Apache License 2.0

[8] https://github.com/Visual-AI/PruneVid, CC BY-NC-SA 4.0 License

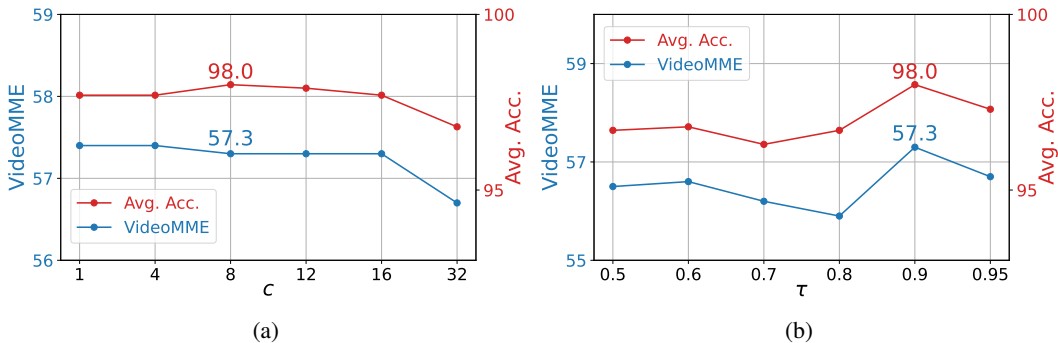

(a)  (b)

Figure 6: Ablation study on $c$ and $\tau$ in DySeg. $c$ denotes the minimum number of segments in a video, whereas $\tau$ denotes the transition similarity threshold. Both parameters jointly control the segmentation granularity.

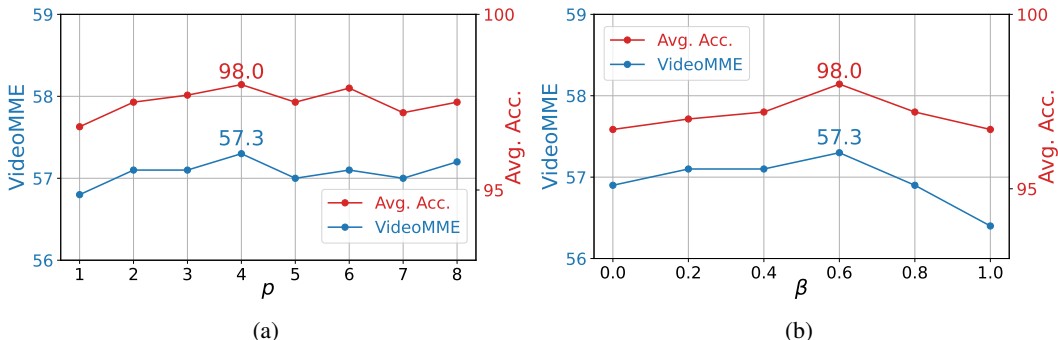

(a)  (b)

Figure 7: Ablation study on $p$ and $\beta$ in DTM. $p$ denotes the interval for anchor frame selection, whereas $\beta$ controls the merging weight for anchor tokens and their associated tokens in Eq. (5).

## B Additional Experimental Results

### B.1 Ablation study on $c$ and $\tau$ in DySeg

We evaluate the effect of $c$ and $\tau$ in Eq. (2) of DySeg. Figure 6(a) shows results with varing $c$. In simple video scenarios with high transition similarity, $c$ regulates segmentation. When $c = 1$, segmentation relies solely on $\tau$. $c = 32$ divides the video into single-frame segments. Performance remains stable for $1 \leq c \leq 16$, but the decline at $c = 32$ suggests that excessive segmentation disregards temporal relationships. The optimal performance at $c = 8$ indicates the benefit of a minimum cluster number. Figure 6(b) shows results with varing $\tau$. When $\tau$ is small, most transitions in $\mathbf{S}$ are selected by $c$. As $\tau$ increases, more transitions fall below the threshold. The best performance is achieved at $\tau = 0.9$, effectively grouping redundant frames while separating non-redundant ones.

### B.2 Ablation study on $p$ and $\beta$ in DTM

In Figure 7(a), we evaluate the effect of $p$ on anchor frame selection, from which anchor tokens are subsequently sampled. When $p = 1$, anchor tokens are evenly distributed across all frames within a segment. As $p$ increases, fewer frames are selected as anchors, while more anchor tokens are allocated to each anchor frame. Given the high similarity between frames in a segment, a sufficient number of anchor tokens per anchor frame is essential to capture visual context. However, if $p$ is too large, all anchor tokens come from the first frame, limiting temporal information. We find that $p = 4$ yields the best performance by effectively capturing spatiotemporal context. Figure 7(b) shows the effect of $\beta$ in Eq. (5). When $\beta = 0.0$, matching tokens are averaged. $\beta = 1.0$ discards all non-anchor tokens, removing segment visual context and leading to a performance drop. Notably, Anchor-centric Assignment (when $\beta = 0.6$) yields optimal results, highlighting the importance of representative tokens and visual context.

Table 11: Additional quantitative results on VideoMME. We report the overall performance of LLaVA-OneVision.

| Method | Retention Ratio $R$ | TFLOPs | VideoMME |
|---|---|---|---|
| Vanilla | 100% | 48.82 | 58.6 |
| PyramidDrop [47]CVPR'25 | 100%/4.1%/2.0%/1.0% | 14.70 | 51.8 |
| SparseVLM [56]ICML'25 | 100%/10.4%/5.3%/2.9% | 7.06 | 53.3 |
| LLaVA-PruMerge [32]ICCV'25 | 9.7% | 4.04 | 49.9 |
| **FastVID** | 9,7% | 4.04 | **57.3** |

Table 12: Additional quantitative results on ANet-QA. We report the performance of LLaVA-OneVision.

| Method | TFLOPs | ANet-QA | |
|---|---|---|---|
| | | Accuracy | Score |
| Vanilla | 48.82 | 54.4 | 3.57 |
| PyramidDrop | 14.70 | 41.2 | 3.07 |
| **FastVID** | 4.04 | **53.1** | **3.52** |

## B.3 Additional Quantitative Results

In Table 11, we conduct additional comparative experiments against previous methods.

**Comparison with PyramidDrop** [47]. Following its official settings, we prune tokens at the 8th, 16th, and 24th layers with retention ratios of 4.1%, 2.0%, and 1.0%, respectively. Despite aggressive pruning, PyramidDrop still incurs high FLOPs due to full video token processing in the first 8 layers. In contrast, our FastVID compresses tokens from the input stage, reduces computation significantly, and outperforms PyramidDrop under extreme pruning.

**Comparison with SparseVLM** [56]. Following its official setup, we prune tokens at the 3rd, 7th, and 16th layers with retention ratios of 10.4%, 5.3%, and 2.9%, respectively. Although SparseVLM aggressively prunes later layers, it still incurs high FLOPs due to full token processing in the early layers. In contrast, FastVID compresses tokens from the input stage, leading to significantly lower computational cost. Moreover, FastVID achieves better performance due to its more effective video token pruning strategy.

**Comparison with LLaVA-PruMerge** [32]. Although both LLaVA-Prumerge and our ATS select tokens based on attention scores, our FastVID is specifically designed to better preserve both temporal and visual context in Video LLMs. A direct quantitative comparison is provided in Table 11. Under the same video token budget of 608 tokens ($\approx$9.7% retention ratio), our method significantly outperforms LLaVA-PruMerge.

## B.4 Results for long-context generation

FastVID prunes redundant video tokens while preserving critical semantic information, which supports both short- and long-context generation tasks. We select ANet-QA [52] for the video captioning task reported in Table 12. Metrics are computed using GPT-3.5-Turbo-0125. FastVID achieves **97.6%** accuracy and **98.6%** score, demonstrating its effectiveness in long-context generation.

## B.5 Additional Qualitative Results

Figures 8-10 show additional qualitative results on VideoMME. We use the following settings: $c = 0, \tau = 0.9, d = 0.4, p = 2$, retaining an average of 10 tokens per frame. Figure 8 presents static scenes, while Figures 9 and 10 present dynamic scenes. These visualizations demonstrate FastVID's ability to adaptively segment videos across varying temporal dynamics. Within each segment, the merged region boundaries loosely align with object shapes, highlighting the effectiveness of the proposed DTM in preserving visual context.

Figures 8(b) and 9(a) showcase failure cases where the query is relevant to only a small subset of frames. In Figure 8(b), only a few frames depict the subject eating a banana, while in Figure 9(a), only a few frames correspond to the entrance scene. In such scenarios, when detailed questions are posed, our query-agnostic pruning strategy becomes less effective. Under high compression rates, most retained tokens are allocated to query-irrelevant frames, making it challenging for the model to provide the necessary visual evidence and produce accurate responses.

# C  Additional Discussions

## C.1  Comparison with PruneVID

PruneVID [16] and FastVID differ fundamentally in both design and implementation. Below, we detail the key technical distinctions and their practical impact on efficiency and accuracy.

1. Methodological Differences
   (a) Segmentation Strategy: PruneVID uses the original density-based clustering to partition the video into segments, with a hard upper bound on segment number ($\leq$ sampled frames $\times \gamma$ = 0.25). **This constraint may cause semantically diverse frames to be grouped together in complex videos**, reducing the effectiveness of subsequent pruning. In contrast, FastVID uses frame transition similarity to adaptively segment the video, **ensuring high intra-segment similarity**, thereby facilitating more efficient and accurate token pruning.
   (b) Compression Strategy: PruneVID performs clustering-based merging (Figure 4(a)) on both dynamic and static tokens within each segment, **without considering token representativeness or positional structure**. In contrast, our proposed density-based token merging DTM selects anchor tokens only from anchor frames and merges the remaining tokens within each segment. The number of anchor tokens per frame is adaptive to segment length. Importantly, **our DTM explicitly emphasizes anchor tokens located at density peaks**. Positional information of anchor tokens is preserved to maintain the spatiotemporal structure, which is particularly beneficial for the Qwen-VL series that adopts M-RoPE. We further apply anchor-centric aggregation instead of simple average pooling to highlight representative anchor tokens.

2. Practical Impact
   (a) Efficiency: In Table 5, we conduct a detailed efficiency comparison. The primary bottleneck in efficiency lies in the density score computation. Our segmentation based on frame transition similarity is significantly faster than PruneVID's clustering. During compression, PruneVID clusters dynamic and static tokens separately per segment. Crucially, the number of tokens varies across segments, which leads to repeated density computations. In contrast, we computes density scores in parallel across all anchor frames, requiring only a single pass. As a result, our pruning speed is **6.1$\times$** faster (**6.1ms** vs. 37.2ms).
   (b) Accuracy: We compare PruneVID across multiple Video LLMs, including LLaVA-OneVision (Table 1), Qwen2-VL (Table 3), and Qwen2.5-VL (Table 4). Under both the 32 fixed-frame sampling setting (LLaVA-OneVision) and dynamic frame sampling (Qwen-VL series), our method consistently outperforms PruneVID at the similar compression ratio. We attribute this improvement to our more effective dynamic segmentation strategy, which ensures high intra-segment similarity. Furthermore, our token merging strategy highlights representative tokens while retaining thier positional information, making it well-suited for diverse models.

In summary, FastVID introduces a novel framework that integrates dynamic temporal segmentation with density spatiotemporal pruning, effectively preserving both temporal and visual context. It achieves consistent improvements in both efficiency and accuracy over prior methods across diverse Video LLMs.

## C.2  Limitations

FastVID achieves strong performance on LLaVA-OneVision [19] (**32** frames/video), maintaining comparable accuracy with only **8.3%** of the FLOPs. However, on LLaVA-Video [57] (**64** frames/video), Qwen2-VL [45] (**768** frames/video) and Qwen2.5-VL [4] (**768** frames/video), although FastVID consistently outperforms existing SOTA baselines, a noticeable accuracy drop occurs compared to the original model.

This degradation mainly arises from the nature of our query-agnostic pruning strategy. Our aggressive pruning retains only a small number of tokens. When temporal spans are long, few of these retained tokens relate to the query, leading to performance drops. To overcome these challenges, future work may explore integrating query-guided keyframe selection to better support long-frame Video LLMs [45, 57].

However, query-agnostic pruning has a key advantage: the pruned KV cache can be reused across multiple dialogue turns with different queries, making it more suitable for multi-turn conversations.

Both query-aware and query-agnostic methods have their own advantages, and the choice can be made based on specific application scenarios.

### C.3 Broader Impacts

FastVID accelerates inference for existing Video LLMs without modifying their parameters or training new models, thereby minimizing the risk of introducing new biases or unintended behaviors. However, FastVID inherits any potential negative societal impacts of the original models, such as representational bias or potential misuse. Despite this, FastVID maintains strong performance even under extreme token compression. This makes the use of Video LLMs more practical in environments with limited computational resources, promoting broader and more sustainable deployment.

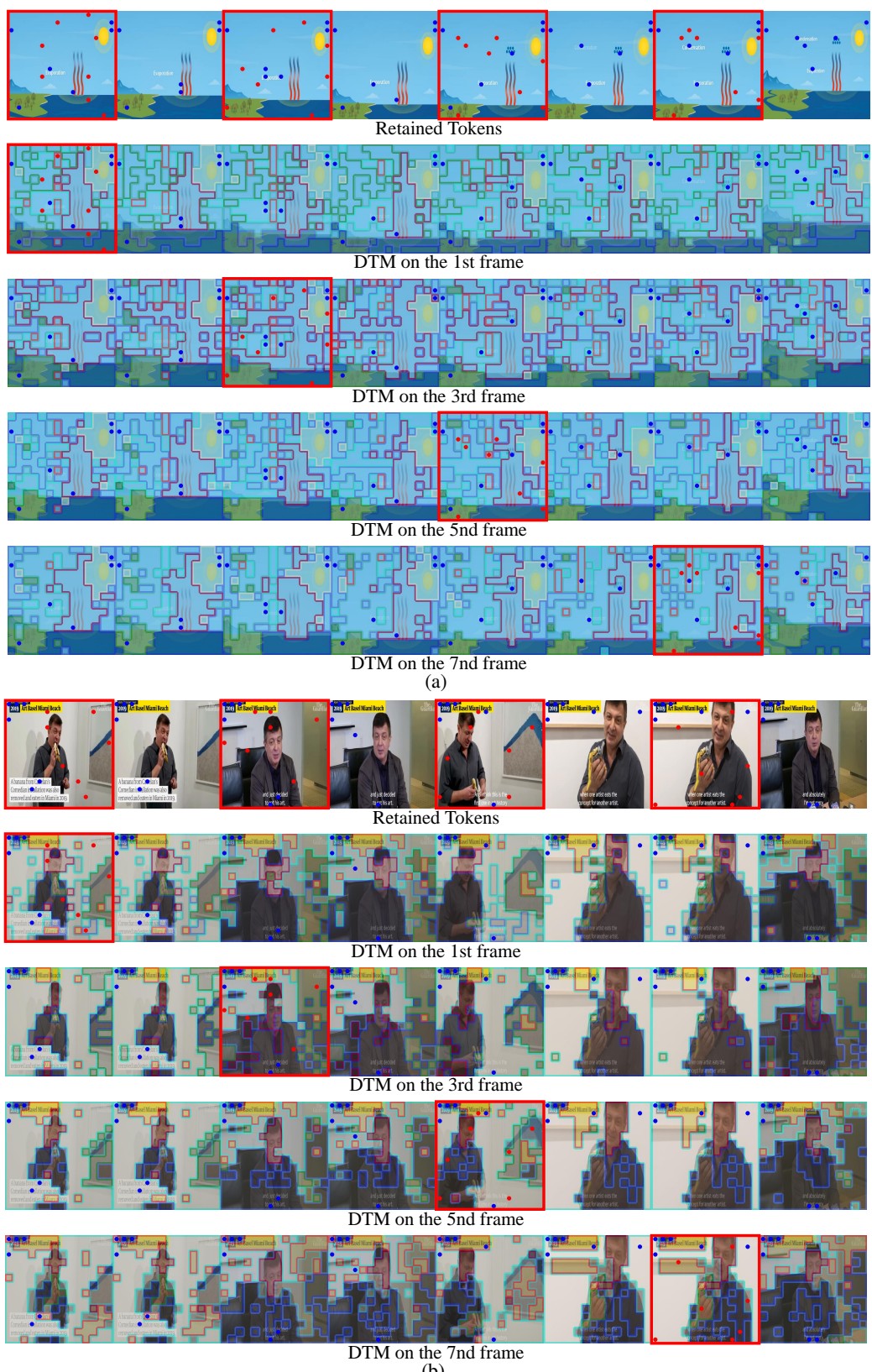

Figure 8: Segment boundaries are marked by brown vertical lines. Tokens generated by DTM and ATS are highlighted in red and blue, respectively. Anchor frames, indicated by red boxes, are processed by DTM individually. In DTM, patches with matching inner and border colors are merged.

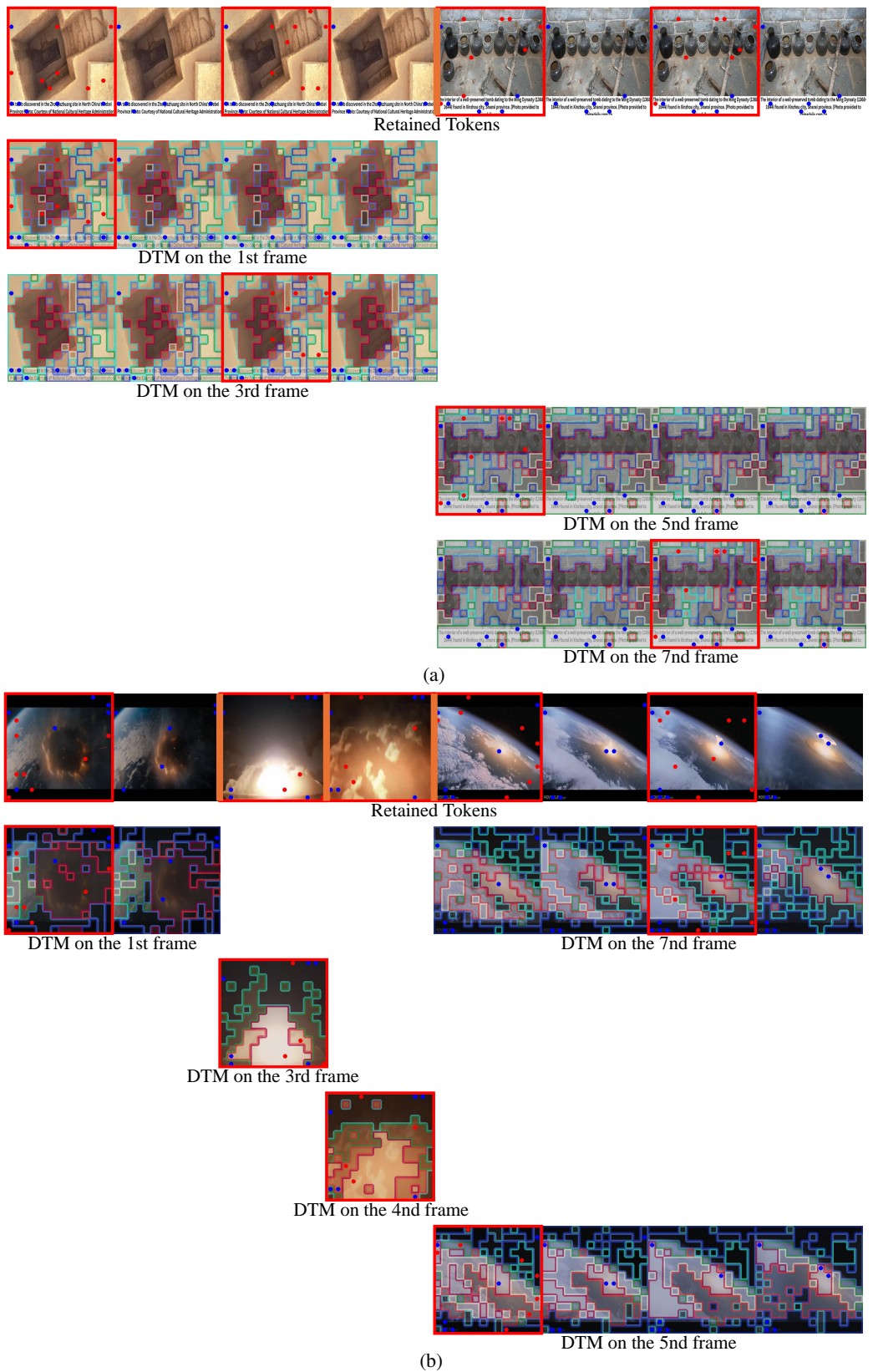

Figure 9: Segment boundaries are marked by brown vertical lines. Tokens generated by DTM and ATS are highlighted in red and blue, respectively. Anchor frames, indicated by red boxes, are processed by DTM individually. In DTM, patches with matching inner and border colors are merged.

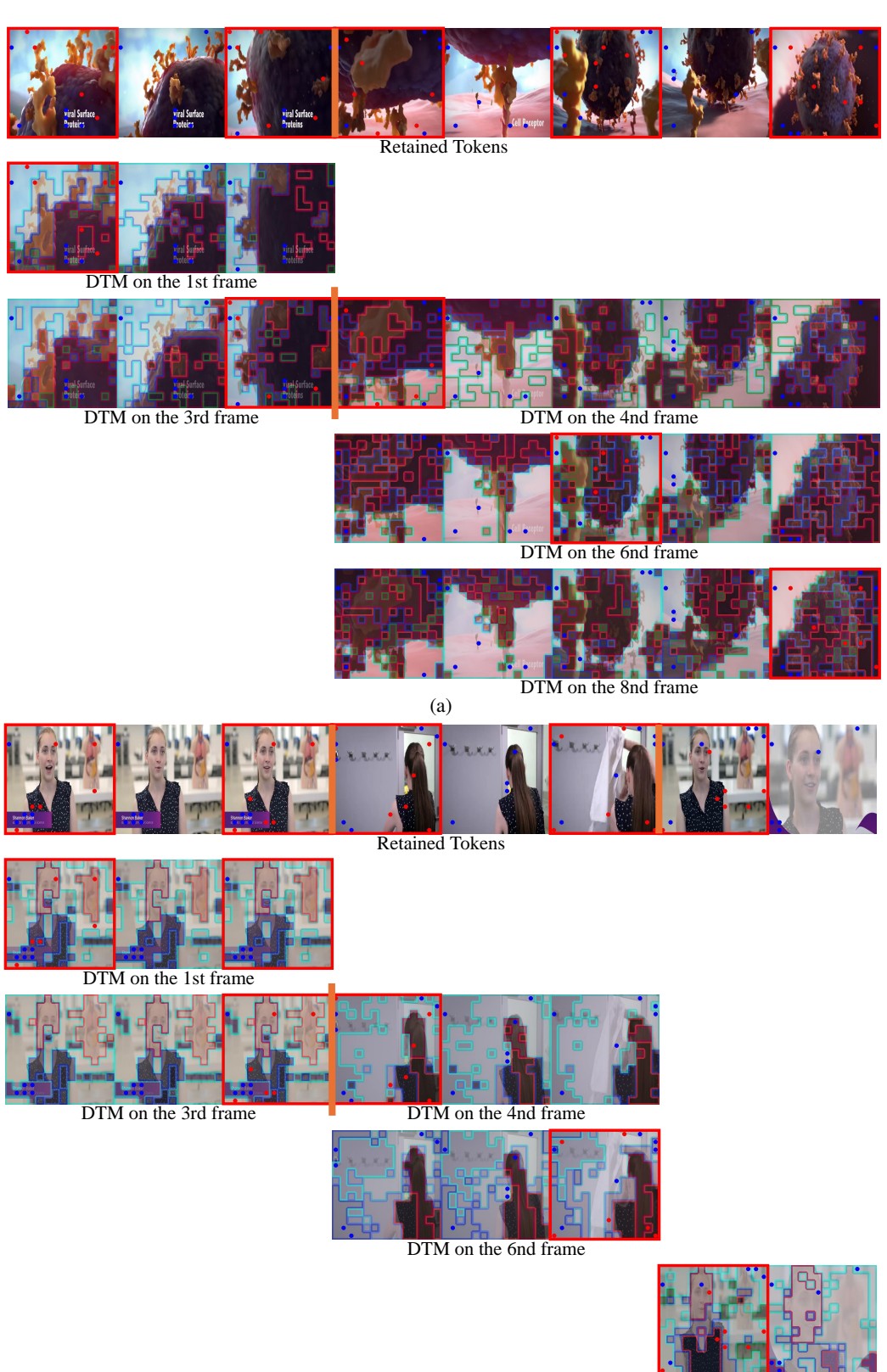

Figure 10: Segment boundaries are marked by brown vertical lines. Tokens generated by DTM and ATS are highlighted in red and blue, respectively. Anchor frames, indicated by red boxes, are processed by DTM individually. In DTM, patches with matching inner and border colors are merged.

