# OpenReview forum: "FastVID: Dynamic Density Pruning for Fast Video Large Language Models"
_NeurIPS.cc/2025/Conference — NeurIPS 2025 poster_

### Official Review · Reviewer_GXf1 · 2025-06-03

**Clarity:** 3
**Significance:** 2
**Originality:** 2
**Rating:** 4
**Confidence:** 4

**Summary:**

This paper proposes a token pruning method called FastVID, which effectively improves the efficiency of video LLMs (LLaVA-OV and LLaVA-Video) across a wide a range of video benchmarks. Specifically, FastVID segments the video into segments and conducts token pruning within each one, achieving superior performance compared to existing methods.

**Questions:**

1. In Tab.2, FastVID achieves significant efficiency improvement compared to the vanilla model. I would like to know which attention version the vanilla model is using.

**Ethical Concerns:**

["NO or VERY MINOR ethics concerns only"]

**Final Justification:**

The authors have addressed my concerns. So, I will keep my positive rating.

**Limitations:**

Yes.

**Quality:**

2

**Strengths And Weaknesses:**

**Strengths**
1. The paper writing is clear, and the method is easy to follow.
2. FastVID achieves SOTA performance compared to previous methods.

**Weaknesses**

1. **Technical Novelty and Comparison with Prior Work:**
   The proposed method appears to combine existing approaches, specifically dynamic temporal segmentation, density-based token merging, and attention-based token selection. From my understanding, the concepts of dynamic temporal segmentation and density-based token merging are already central to PruneVid [1]. The main technical distinction highlighted in this paper is that PruneVid [1] uses clustering to find dynamic anchors for token merging (which introduces additional latency), whereas the current paper uses fixed anchors.
   However, the discussion in the related work section is brief and does not make it clear whether anchor selection is the **only** substantive difference between the proposed method and PruneVid [1] or if there are other key innovations. I would appreciate a more detailed technical comparison, clarifying (i) all methodological differences and (ii) their practical impact on efficiency and accuracy.

   Furthermore, the attention-based token selection in this paper appears very similar to the approach used in LLaVA-Prumerge [2]. While this similarity is acknowledged in the related work section, there is no direct quantitative comparison provided with LLaVA-Prumerge [2]. A side-by-side performance evaluation would be necessary to assess the contribution and effectiveness of the proposed method relative to LLaVA-Prumerge [2].

2. **Comparative Evaluation:**
   Other relevant token pruning methods, such as SparseVLM [3], are only mentioned briefly and not included in the experimental comparison. Incorporating these methods into the evaluation would provide a more comprehensive and convincing assessment of the strengths and weaknesses of the proposed approach.

In summary, while the Related Work section makes some connections to prior work, it does not fully address (a) the precise technical distinctions between the proposed method and PruneVid [1], nor (b) provide sufficient experimental comparison with LLaVA-Prumerge [2] and other relevant methods like SparseVLM [3]. Addressing these points would help clarify the technical novelty and practical value of the paper.

[1] "PruneVid: Visual Token Pruning for Efficient Video Large Language Models." arXiv preprint arXiv:2412.16117 (2024).

[2] "Llava-prumerge: Adaptive token reduction for efficient large multimodal models." arXiv preprint arXiv:2403.15388 (2024).

[3] "Sparsevlm: Visual token sparsification for efficient vision-language model inference." arXiv preprint arXiv:2410.04417 (2024).

---

> ### Author Rebuttal · Authors · 2025-07-31
>
> Thank you for your positive and insightful comments, which has been instrumental in improving the overall quality of our work. In response, we have performed further experiments and analyses. All clarifications and results will be included in the revised paper.
>
> ***
>
> **Question 1**: The precise technical distinctions between the proposed method and PruneVid [1]. I would appreciate a more detailed technical comparison, clarifying (i) all methodological differences and (ii) their practical impact on efficiency and accuracy.
>
> [1] PruneVid: Visual Token Pruning for Efficient Video Large Language Models. ACL25.
>
> **Response**: Thank you for the thoughtful feedback. PruneVID (ACL25) and FastVID differ fundamentally in both design and implementation. Below, we detail the key technical distinctions and their practical impact on efficiency and accuracy.
>
> 1. Methodological Differences
> 	- Segmentation Strategy: PruneVID uses the original density-based clustering to partition the video into segments, with a hard upper bound on segment number ($\leq$ sampled frames $\times$ $\gamma$ = 0.25). **This constraint may cause semantically diverse frames to be grouped together in complex videos**, reducing the effectiveness of subsequent pruning. In contrast, FastVID uses frame transition similarity to adaptively segment the video, **ensuring high intra-segment similarity**, thereby facilitating more efficient and accurate token pruning.
> 	 - Compression Strategy: PruneVID performs clustering-based merging (Fig. 4(a) in the main paper) on both dynamic and static tokens within each segment, **without considering token representativeness or positional structure**. In contrast, our proposed density-based token merging DTM (lines 176–180) selects anchor tokens only from anchor frames and merges the remaining tokens within each segment. The number of anchor tokens per anchor frame is adaptive to segment length, rather than fixed. Importantly, **our DTM explicitly emphasizes anchor tokens located at density peaks** (lines 193-203). Positional information of anchor tokens is preserved to maintain the spatiotemporal structure, which is particularly beneficial for the Qwen-VL series that adopts M-RoPE. We further apply anchor-centric aggregation instead of simple average pooling to highlight representative anchor tokens.
> 2. Practical Impact
> 	- Efficiency: In Table 3 of the main paper, we conduct a detailed efficiency comparison (lines 262–266). The primary bottleneck in efficiency lies in the density score computation. Our segmentation based on frame transition similarity is significantly faster than PruneVID’s clustering. During compression, PruneVID clusters dynamic and static tokens separately per segment. Crucially, the number of tokens varies across segments, which leads to repeated density computations. In contrast, we computes density scores in parallel across all anchor frames, requiring only a single pass. As a result, our pruning speed is **6.1×** faster (**6.1ms** vs. 37.2ms).
> 	- Accuracy: We compare PruneVID across multiple Video LLMs, including LLaVA-OneVision (Table 1 in the main paper), Qwen2-VL (Table 7 in the appendix), and Qwen2.5-VL (response to Reviewer JwZp's Q1). Under both fixed frame sampling (LLaVA-OneVision) and dynamic frame sampling (Qwen-VL series), our FastVID consistently outperforms PruneVID at the similar compression ratio, demonstrating its strong compression capability.
>
> In summary, FastVID introduces a novel framework that integrates dynamic temporal segmentation with density spatiotemporal pruning, effectively preserving both temporal and visual context. It achieves consistent improvements in both efficiency and accuracy over prior methods across diverse Video LLMs.
>
> ***
>
> **Question 2**: The attention-based token selection (ATS) in this paper appears very similar to the approach used in LLaVA-Prumerge [2]. While this similarity is acknowledged in the related work section, there is no direct quantitative comparison provided with LLaVA-Prumerge [2].
>
> [2] Llava-prumerge: Adaptive token reduction for efficient large multimodal models. ICCV25.
>
> **Response**: Thank you for the observation. As stated in line 206, our ATS is indeed inspired by LLaVA-PruMerge (which we cite as [27]). However, our FastVID is specifically designed to better preserve both temporal and visual context in Video LLMs. A direct quantitative comparison is provided in Table 1. Under the same video token budget of 608 tokens ($\approx$ 9.7% retention ratio), our method significantly outperforms LLaVA‑PruMerge.
>
> #### **Table 1: Comparisons with LLaVA-Prumerge and SparseVLM.**
> |LLaVA-OneVision|Retention Ratio|TFLOPs|VideoMME|
> |:---:|:---:|:---:|:---:|
> |Vanilla|100%|48.82|_58.6_|
> |SparseVLM (ICML25)|100%/10.4%/5.3%/2.9%|7.06|53.3|
> |LLaVA-PruMerge (ICCV25)|9.7%|4.04|49.9|
> |FastVID|9.7%|4.04|**57.3**|
>
> ***
>
> **Question 3**: Other relevant token pruning methods, such as SparseVLM [3], are only mentioned briefly and not included in the experimental comparison.
>
> [3] Sparsevlm: Visual token sparsification for efficient vision-language model inference. ICML25.
>
> **Response**: Thank you for the suggestion. We include a direct comparison with SparseVLM in Table 1. Following its official setup, we prune tokens at the 3rd, 7th, and 16th layers with retention ratios of 10.4%, 5.3%, and 2.9%, respectively. Although SparseVLM aggressively prunes later layers, it still incurs high FLOPs due to full token processing in the early layers. In contrast, FastVID compresses tokens from the input stage, leading to significantly lower computational cost. Moreover, FastVID achieves better performance due to its more effective video token pruning strategy.
>
> ***
>
> **Question 4**: FastVID achieves significant efficiency improvement compared to the vanilla model. I would like to know which attention version the vanilla model is using.
>
> **Response**: Thank you for your question. In the efficiency comparison (Table 3 of the main paper), all models, including the vanilla model, use FlashAttention (version 2.7.2.post1) to ensure a fair evaluation.

---

> > ### Comment · Reviewer_GXf1 · 2025-08-01
> > **Response to Authors**
> >
> > I am satisfied with the response and have no more concerns. The authors should include all the discussions and results in their revision.
> >
> > Thanks.

---

> > > ### Author Response · Authors · 2025-08-01
> > > **Official Comment by Authors**
> > >
> > > Thank you for acknowledging that our response has addressed your concerns. We will include all the discussions and results in the revised version. We sincerely appreciate your constructive review.

---

### Official Review · Reviewer_vfkR · 2025-06-09

**Clarity:** 3
**Significance:** 3
**Originality:** 3
**Rating:** 4
**Confidence:** 4

**Summary:**

This paper introduce FastVID, a novel inference-time pruning framework designed to accelerate Video LLMs by effectively reducing spatiotemporal redundancy. It achieves a 7.1× speedup while preserving stable accuracy. However, more models, benchmarks and prior works should be considered.

**Questions:**

please refer to weaknesses

**Ethical Concerns:**

["NO or VERY MINOR ethics concerns only"]

**Final Justification:**

Thanks for your rebuttal, my concerns are resolved and I will maintain my score. I suggest authors include the new discussions and experiments in their revision.

**Limitations:**

yes

**Quality:**

3

**Strengths And Weaknesses:**

Strengths

1. The paper is well-organized.

2. The figures are visually good and easy to understanding.

3. The method solves the problem well. It efficiently prunes the redundant visual tokens in video and achieves 7x acceleration for video LLM.

Weaknesses

1. More video LLMs should be considered in experiments (e.g. Qwen2.5-vl). LLava-onevision and qwen2-vl employ the same LLM (qwen2), it is better for authors that testing their method in other MLLMs based on various LLM.

2. The datasets are almost VideoQA, what about the performance on other long context generation datasets? Because the method proposed also prune KV Cache, will it cause a decrease in quality on long context  generation(like video captioning task ActivityNet, VideoDetailCaption e.g.)

3. Some prior works VTW[1], ST3[2], pyramiddrop[3] should be discussed in related work or compared in experiments:
The FastV is a old baseline, so the recent method blow should be consider in comparison.

[1] Boosting Multimodal Large Language Models with Visual Tokens Withdrawal for Rapid Inference. AAAI25

[2] ST3: Accelerating Multimodal Large Language Model by Spatial-Temporal Visual Token Trimming. AAAI25

[3] PyramidDrop: Accelerating Your Large Vision-Language Models via Pyramid Visual Redundancy Reduction. CVPR25

---

> ### Author Rebuttal · Authors · 2025-07-31
>
> Thank you for your positive and insightful comments, which has been instrumental in improving the overall quality of our work. In response, we have performed further experiments and analyses. All clarifications and results will be included in the revised paper.
>
> ***
>
> **Question 1**: More video LLMs should be considered in experiments (e.g.  Qwen2.5-vl). LLava-onevision and qwen2-vl employ the same LLM (qwen2), it is better for authors that testing their method in other MLLMs based on various LLM.
>
> **Response**: Thank you for the valuable suggestion. To evaluate the generalization ability of FastVID, we report results on Qwen2.5-VL in Table 1. Unlike the LLaVA series and Qwen2-VL, Qwen2.5-VL employs a vision encoder with window attention and a Qwen2.5 LLM, representing a substantially different architecture. Compared to the recent SOTA PruneVID (ACL25), FastVID achieves a **3.5** gain under a comparable retention ratio. This supports that FastVID is model-agnostic and generalizes well across diverse Video LLM architectures.
>
> #### **Table 1: Results on Qwen2.5-VL.**
> |Qwen2.5-VL|Retention Ratio|TFLOPs|VideoMME|
> |:---:|:---:|:---:|:---:|
> |Vanilla|100%|124|_67.3_|
> |PruneVID* (ACL25)|24.5%|23.7|59.3|
> |FastVID|24.1%|23.2|**62.8**|
>
> ***
>
> **Question 2**: The datasets are almost VideoQA, what about the performance on other long context generation datasets? Because the method proposed also prune KV Cache, will it cause a decrease in quality on long context generation(like video captioning task ActivityNet,  VideoDetailCaption e.g.)
>
> **Response**: Thank you for the insightful question. FastVID prunes redundant video tokens while preserving critical semantic information, which supports both short- and long-context generation tasks. Due to time constraints, we select ActivityNet for the video captioning task reported in Table 2. Metrics are computed using GPT-3.5-Turbo-0125. FastVID achieves **97.6%** accuracy and **98.6%** score, demonstrating its effectiveness in long-context generation.
>
> #### **Table 2: Comparisons with PyramidDrop on VideoMME and ActivityNet.**
> |LLaVA-OneVision|Retention Ratio|TFLOPs|VideoMME|ActivityNet||
> |:---:|:---:|:---:|:---:|:---:|:---:|
> ||||Overall|Accuracy|Score|
> |Vanilla|100%|48.82|_58.6_|_54.4_|_3.57_|
> |PyramidDrop (CVPR25)|100%/4.1%/2.0%/1.0%|14.70|51.8|41.2|3.07|
> |FastVID|9.7%|4.04|**57.3**|**53.1**|**3.52**|
>
> ***
>
> **Question 3**: Some prior works VTW[1], ST3[2], pyramiddrop[3] should be discussed in related work or compared in experiments. The FastV is a old baseline, so the recent method blow should be consider in comparison.
>
> [1] Boosting Multimodal Large Language Models with Visual Tokens Withdrawal for Rapid Inference. AAAI25
>
> [2] ST3: Accelerating Multimodal Large Language Model by Spatial-Temporal Visual Token Trimming. AAAI25
>
> [3] PyramidDrop: Accelerating Your Large Vision-Language Models via Pyramid Visual Redundancy Reduction. CVPR25
>
> **Response**: Thank you for highlighting these relevant works. In addition to FastV, we already compare against several recent SOTA baselines, including VisionZip (CVPR25), DyCoke (CVPR25), and PruneVID (ACL25). We will include VTW[1], ST3[2], and PyramidDrop[3] in the revised version of our paper.
>
> VTW removes visual tokens at a specific layer, arguing they are redundant in deep layers of MLLMs. ST3 dynamically adjust the number of visual tokens across LLM layers and generation steps. PyramidDrop performs multi-stage visual token dropping to improve efficiency. However, none of these methods consider temporal relationships across video frames, which are critical for Video LLMs.
>
> Due to time constraints, we include PyramidDrop as a baseline in Table 2. Following its official settings, we prune tokens at the 8th, 16th, and 24th layers with retention ratios of 4.1%, 2.0%, and 1.0%, respectively. Despite aggressive pruning, PyramidDrop still incurs high FLOPs due to full video token processing in the first 8 layers. In contrast, our FastVID compresses tokens from the input stage, reduces computation significantly, and outperforms PyramidDrop under extreme pruning on both VideoMME and ActivityNet.

---

> ### Comment · Reviewer_vfkR · 2025-08-05
>
> Thanks for your rebuttal, my concerns are resolved and I will maintain my score. I suggest authors include the new discussions and experiments in their revision.

---

> > ### Author Response · Authors · 2025-08-05
> > **Official Comment by Authors**
> >
> > Thank you for your suggestion. We will include the new discussions and experiments in the revised version. We are glad that your concerns have been addressed.

---

### Official Review · Reviewer_QvJS · 2025-07-03

**Clarity:** 3
**Significance:** 2
**Originality:** 2
**Rating:** 4
**Confidence:** 3

**Summary:**

This paper introduces FastVID, a dynamic token pruning method designed to reduce the cost of Video LLMs by addressing spatiotemporal redundancy. FastVID analyzes video redundancy through temporal and visual contexts, then applies a two-part strategy: (1) temporal segmentation to preserve sequence structure, and (2) density-based pruning to retain essential visual tokens. Evaluated on models like LLaVA-OneVision, FastVID achieves state-of-the-art efficiency-performance tradeoffs across various video benchmarks.

**Questions:**

- (maybe I missed sth. somewhere) in line 225: why adopts two representative Video LLMs? what is the tradeoff for each base model?
(Thanks the author for explaining the reason.)

**Ethical Concerns:**

["NO or VERY MINOR ethics concerns only"]

**Final Justification:**

All my concerns are addressed: eg, 1, revision the writing to be more objective on arguments and (no more strong over claim); 2, explained reasons of adopting two baselines. Will keep the score unchanged given the impact of the main subject of this paper.

**Limitations:**

yes

**Quality:**

2

**Strengths And Weaknesses:**

Strengths:
- focus on efficiency of video processing, a key bottleneck of video LLMs;
- strong tradeoff on performance vs efficiency (98% of accuracy with 90.3% pruned tokens and 8.3% of FLOPs);
- figures are intuitive to understand the benefits of the proposed method;
- diverse benchmarks and detailed ablation studies;

Weaknesses:
- some arguments are too strong: i) in abstract, “fully exploit”: probably no work(includes this paper) can do fully but only improve? ii) line 26: “nature of video redundancy”, there’s probably no such nature exists, video redundancy may just be about having high-frequency changes that human does not care much at this stage? iii) line 134: “without loss of key information”, without is too strong (the definition of key information may vary and there’s always a loss of information for any neural network?)
- general presentation issue: i) in abstract etc. “visual context”: visual is very general and covers temporal, here it probably means spatial; ii) Figure 1 (b), (b1) and (b2) are not easy to understand at the beginning, probably because lacking description/legend about what does purple/green etc. area mean? iii) Sec. 3.1 sounds like a tone of another abs/intro.
- limited novelty, compress video tokens/making efficiency tradeoff is very common in video LLMs.

---

> ### Author Rebuttal · Authors · 2025-07-31
>
> Thank you for your positive and insightful comments, which has been instrumental in improving the overall quality of our work. In response, we have performed further analyses. All clarifications will be included in the revised paper.
>
> ***
>
> **Question 1**: Some arguments are too strong. 1. "Fully exploit". 2. "Nature of video redundancy". 3. "Without loss of key information".
>
> **Response**: Thank you for your detailed suggestions regarding our wording. We agree that some phrases can be overly strong and will revise them accordingly. Specifically:
>
>  1. "Existing pruning techniques fail to _effectively exploit_ the spatiotemporal redundancy present in video data."
>  2. "As a result, _spatiotemporal redundancy_ remain insufficiently explored."
>  3. "..., enhancing efficiency _with minimal loss_ of key information."
>
> Additionally, we will revise the full manuscript to improve clarity and precision, aided by writing tools and consulting native speakers.
>
> ***
>
> **Question 2**: General presentation issue. 1. "Visual context". 2. Figure 1(b) lacks descriptions. 3. Sec. 3.1 sounds like a tone of another abs/intro.
>
> **Response**: Thank you for the detailed and constructive suggestions.
>
>  1. We refer to _visual context_ as the both spatial and temporal information within each video segment. After partitioning the video into temporally ordered, high-redundancy segments, we propose density spatiotemporal pruning to preserve the _visual context_ within each segment.
>  2. We will revise both the caption and the legend in Figure 1 to improve clarity. Specifically, we will emphasize that regions with the same color are merged together. For example, the purple area indicates that patches from the hand region are merged.
>  3. In Sec. 3.1, we will remove lines 111–121 and revise lines 122–135 to focus solely on introducing the framework of FastVid, ensuring it reads as a technical section rather than a second abstract.
>
> ***
>
> **Question 3**: limited novelty, compress video tokens/making efficiency tradeoff is very common in video LLMs.
>
> **Response**: Thank you for your comment. While token compression is indeed a common goal in Video LLMs, we highlight that our method introduces a novel framework that combines dynamic temporal segmentation with density spatiotemporal pruning to preserve both temporal and visual context.
>
> Most prior methods (e.g., Dycoke CVPR25, PruneVID ACL25) rely on fixed-length segmentation or clustering with global constraints, which often group semantically diverse frames into the same segment. This leads to pruning important context or merging visually distinct tokens. In contrast, our FastVID adaptively identifies highly redundant segments and preserves representative and distinctive tokens, enabling efficient yet accurate pruning. Our FastVID significantly improves both efficiency and accuracy over prior methods across diverse Video LLMs.
>
> ***
>
> **Question 4.1**: Why adopts two representative Video LLMs?
>
> **Response**: Thank you for your question. We apply FastVID to LLaVA-OneVision (Table 1 in the main paper), LLaVA-Video (Table 2 in the main paper), Qwen2-VL (Table 7 in the appendix), and Qwen2.5-VL (response to Reviewer JwZp's Q1). These models are selected to demonstrate that our method is model-agnostic and robust across different architectures. FastVID consistently achieves SOTA compression performance on all of them, confirming its general applicability and effectiveness.
>
> ***
>
> **Question 4.2**: What is the tradeoff for each base model?
>
> **Response**: Thank you for your question. The LLaVA series serves as commonly used academic baselines, while the Qwen-VL series comprises powerful open-source MLLMs. Each base model is widely used in the Video LLM community and exhibits distinct designs. LLaVA-Video inserts newline tokens after each height-wise position within each frame. The Qwen-VL series adopts a dynamic resolution and frame sampling strategy, combined with M-RoPE to better handle video tokens. Unlike Qwen2-VL, Qwen2.5-VL employs a vision encoder with window attention and a Qwen2.5 LLM. Our FastVID demonstrates strong generalization and effectiveness across all these mainstream models, validating its robustness under diverse model designs.

---

> > ### Comment · Reviewer_QvJS · 2025-08-04
> >
> > The authors address all my concerns and thanks for the rebuttal.

---

> > > ### Author Response · Authors · 2025-08-05
> > > **Official Comment by Authors**
> > >
> > > We sincerely thank you for your time and detailed review. We are glad that all your concerns have been addressed.

---

### Official Review · Reviewer_JwZp · 2025-07-03

**Clarity:** 3
**Significance:** 3
**Originality:** 3
**Rating:** 5
**Confidence:** 4

**Summary:**

This work addresses the significant computational cost of inference in Video LLMs, which is primarily caused by the large number of redundant video tokens. This work proposes FastVID, an inference-time, plug-and-play token pruning framework that requires no retraining. The core idea is to systematically reduce spatiotemporal redundancy while preserving essential information. FastVID consists of two main stages: (1) Dynamic Temporal Segmentation (DySeg), which adaptively partitions the video into temporally ordered segments of high visual similarity, and (2) Density Spatiotemporal Pruning (STPrune), which operates within each segment to prune tokens. STPrune consists of two modules: Density-based Token Merging (DTM) to retain representative visual context and Attention-based Token Selection (ATS) to capture salient details. Extensive experiments are conducted on two Video LLMs (LLaVA-OneVision and LLaVA-Video) across several challenging benchmarks. The results demonstrate that FastVID can prune over 90% of video tokens, leading to a 7.1x speedup in the prefilling stage, while maintaining 98.0% of the original model's accuracy, significantly outperforming existing token compression methods.

**Questions:**

1.  Experiments on the LLaVA series models are convincing. However, since these models share a similar architectural backbone, demonstrating its effectiveness on a more diverse set of models would significantly improve the claims of generalizability.

2. The current experiments are conducted on inputs of up to 64 frames. Would FastVID still work when scaled to much longer video inputs (e.g., hundreds of frames)? Furthermore, could your method potentially aid in length extrapolation at inference time, allowing models trained on shorter contexts to process longer videos more effectively by compressing the input sequence?

3. The ablation study thoroughly investigates the impact of the parameter d. Could you provide a similar analysis or discussion on the sensitivity of the other key hyperparameters?

4. While the overall performance gap is small, any pruning method inevitably has limitations. Could you provide some qualitative analysis of failure cases? For instance, what types of videos or questions cause FastVID to underperform compared to the vanilla model?

**Ethical Concerns:**

["NO or VERY MINOR ethics concerns only"]

**Final Justification:**

The authors address all my concerns, including new comparison on Qwen2.5-VL, length extrapolation and other ablations. I will raise my rating. And I suggest the authors should include the new discussion proposed in the rebuttal.

**Limitations:**

yes

**Paper Formatting Concerns:**

No Paper Formatting Concerns

**Quality:**

3

**Strengths And Weaknesses:**

**Strengths**
1. This method recognizes the non-uniform distribution of information across the temporal dimension of videos. The proposed Dynamic Temporal Segmentation algorithm is a well-designed for this insight.

2. The authors evaluate FastVID on a wide array of benchmarks that cover short, long, and complex videos.

3. The comparison includes recent and relevant state-of-the-art methods (DyCoke, VisionZip, PruneVID). This work re-implements baselines (e.g., VisionZip*) to ensure a fair comparison (pruning after pooling), which adds to the credibility of the results.

4. The reported results—achieving a massive 7.1x speedup with only a 2% drop in accuracy—are highly impressive.

**Weaknesses**

1. While the authors evaluate their method on two Video LLMs (LLaVA-OneVision and LLaVA-Video), these models share a nearly identical architecture, with the primary difference being the training data. To more convincingly demonstrate the generalization capability of FastVID, it would be beneficial to test it on a strong, open-source Video LLM with a more distinct architecture, such as Qwen2.5-VL. This would provide stronger evidence that the method is truly model-agnostic.

2. The experiments in the paper use a maximum of 64 sampled frames. This input length may not be sufficient to fully evaluate the method's ability to compress temporal redundancy in long videos. The effectiveness of temporal segmentation and pruning might change as the temporal context becomes significantly more extended and complex.

3. The proposed method introduces five hyperparameters, yet the ablation studies primarily focus on d. A sensitivity analysis for the other key parameters is missing. For instance, it is unclear how robust the performance is to changes in the segmentation threshold τ or the anchor sampling interval p. Furthermore, the paper could explore whether some hyperparameters are avoidable. For example, could the merging of anchor and associated tokens be simplified by using a standard pooling operation instead of introducing the balance parameter α?

4. The pruning process in FastVID is content-agnostic, meaning it does not leverage information from the user's instruction or query. This could be a limitation, as tokens relevant to the query might be pruned away if they are not considered salient by the general-purpose [CLS] attention or density-based merging.

5. The training of both LLaVA-Video and LLaVA-OneVision involves fine-tuning the pretrained SigLIP encoder. However, for ATS, the proposed training-free method reintroduces the original, un-finetuned SigLIP head to generate attention scores. This creates a potential parameter gap between the vision encoder used by the Video LLM and the head used for token selection. This discrepancy might have a subtle but non-negligible impact on performance, as the attention scores may not perfectly align with the feature space the fine-tuned model has adapted to. A discussion of this potential issue would strengthen the paper.

---

> ### Author Rebuttal · Authors · 2025-07-31
>
> Thank you for your positive and insightful comments, which has been instrumental in improving the overall quality of our work. In response, we have performed further experiments and analyses. All clarifications and results will be included in the revised paper.
>
> ***
>
> **Question 1**: Experiments on the LLaVA series models are convincing. It would be beneficial to test it on a strong, open-source Video LLM with a more distinct architecture, such as Qwen2.5-VL.
>
> **Response**: Thank you for the insightful suggestion. To further demonstrate its generalization ability, we now include results on Qwen2.5-VL in Table 1. Unlike the LLaVA series and Qwen2-VL (Table 7 of the appendix), Qwen2.5-VL employs a vision encoder with window attention and a Qwen2.5 LLM, representing a substantially different architecture. Compared to the recent SOTA PruneVID (ACL25), FastVID achieves a **3.5** improvement under a comparable retention ratio. This supports that FastVID is model-agnostic and generalizes well across diverse Video LLM architectures.
>
> #### **Table 1: Results on Qwen2.5-VL.**
> |Qwen2.5-VL|Retention Ratio|TFLOPs|VideoMME|
> |:---:|:---:|:---:|:---:|
> |Vanilla|100%|124.0|_67.3_|
> |PruneVID* (ACL25)|24.5%|23.7|59.3|
> |FastVID|24.1%|23.2|**62.8**|
>
> ***
>
> **Question 2**: The current experiments are conducted on inputs of up to 64 frames. Would FastVID still work when scaled to much longer video inputs (e.g., hundreds of frames)?
>
> **Response**:  Thank you for raising this important point. FastVID is fully compatible with long video inputs. In fact, both Qwen2-VL (Table 7 in the appendix) and Qwen2.5-VL (response to Q1) adopt a dynamic resolution and frame sampling strategy, allowing up to 768 sampled frames per video. Specifically, on VideoMME, Qwen2.5-VL processes an average of 549.8 sampeld frames per video. FastVID achieves SOTA performance on both Qwen2-VL and Qwen2.5-VL, demonstrating its effectiveness in handling long and complex temporal contexts.
>
> ***
>
> **Question 3**: Could your method potentially aid in length extrapolation at inference time, allowing models trained on shorter contexts to process longer videos more effectively by compressing the input sequence?
>
> **Response**: Thank you for the insightful question. As shown in Table 2, by applying FastVID with different compression ratios, we extend the number of sampled frames from 32 to 320 while maintaining a comparable total number of input video tokens. This consistently improves performance, which demonstrates that FastVID not only compresses video tokens efficiently but also enables effective length extrapolation at inference time.
>
> #### **Table 2: Improved length extrapolation via FastVID.**
> |LLaVA-OneVision|# Frame|# Token|VideoMME|
> |:---:|:---:|:---:|:---:|
> |Vanilla|32|6272|58.6|
> |FastVID (r=25%)|128|6272|60.4|
> |FastVID (r=10%)|320|6080|**61.4**|
>
> ***
>
> **Question 4**: The ablation study thoroughly investigates the impact of the parameter d. Could you provide a similar analysis or discussion on the sensitivity of the other key hyperparameters?
>
> **Response**: We appreciate your concern about hyperparameter sensitivity. Due to space limits, we report comprehensive analyses of other hyperparameters in the appendix (Sections B.1, B.2; Figures 6, 7). We apologize for the typo in Section B.2. The symbol $\beta$ refers to the balance parameter $\alpha$ in Eq. (5). Specifically, when $\alpha=0$, the merging degenerates into a standard pooling operation (line 843), addressing your suggestion for a simpler alternative.
>
> ***
>
> **Question 5**: This creates a potential parameter gap between the vision encoder used by the Video LLM and the head used for token selection. A discussion of this potential issue would strengthen the paper.
>
> **Response**: Thank you for the insightful comment. We acknowledge the potential parameter gap between the fine-tuned SigLIP encoder and the original SigLIP head. However, we believe this gap has limited impact for the following reasons.
>
> First, the SigLIP head is pretrained jointly with the encoder on large-scale vision-language data, enabling strong generalization and transferability. Second, since the first-stage training of Video LLMs also aims to align visual and textual representations, the SigLIP head is naturally compatible with this alignment objective. Finally, as shown in Table 3, results show that using the SigLIP head leads to a 1.0 improvement, indicating its effectiveness. This is likely because the SigLIP head is explicitly trained to aggregate patch tokens for semantic alignment, making its attention scores better suited for identifying semantic salient tokens.
>
> #### **Table 3: Ablation study on the SigLIP head.**
> |LLaVA-OneVision|VideoMME|
> |:---:|:---:|
> |FastVID|**57.3**|
> |w/o the SigLIP head|56.3|
>
> In addition, we emphasize that our FastVID generalizes to vision encoders without pretrained heads or [CLS] tokens (e.g., Qwen-VL series). In such cases, [CLS] attention scores are computed at the final layer using pseudo [CLS] tokens derived by averaging patch tokens. Our FastVID achieves strong performance on Qwen2-VL (Table 7 in the appendix) and Qwen2.5-VL (response to Q1), significantly outperforming previous methods.
>
> ***
>
> **Question 6.1**: The pruning process in FastVID is content-agnostic. This could be a limitation, as tokens relevant to the query might be pruned away.
>
> **Response**: Thank you for pointing out the limitation of query content-agnostic pruning. Our aggressive pruning retains only a small number of tokens. When temporal spans are long, few of these retained tokens relate to the query, leading to performance drops (discussed in Section C.1 Limitations). However, query-agnostic pruning has a key advantage: the pruned KV cache can be reused across multiple dialogue turns with different queries, making it more suitable for multi-turn interactions. Both query-aware and query-agnostic methods have their own advantages, and the choice can be made based on specific application scenarios.
>
> ***
>
> **Question 6.2**: Could you provide some qualitative analysis of failure cases? For instance, what types of videos or questions cause FastVID to underperform compared to the vanilla model?
>
> **Response**: Thank you for the suggestion. As discussed in our response to Q6.1, we analyze a typical failure case of our FastVID, particularly when the query relates to only a few frames. In such cases, most of the retained tokens after pruning may not be related to the query. Unfortunately, we are unable to include qualitative examples during the rebuttal phase due to formatting constraints, but we will provide detailed failure cases in the revised version.

---

> ### Comment · Reviewer_JwZp · 2025-08-07
>
> Thanks for your rebuttal. The authors address all my concerns and I will keep my rating. I suggest the authors should include the new discussion proposed in the rebuttal.

---

> > ### Author Response · Authors · 2025-08-08
> > **Official Comment by Authors**
> >
> > Thank you for your valuable feedback. We are pleased to hear that all your concerns have been addressed. As suggested, we will include the new discussions in the revised version.

---

### Note · Authors · 2025-08-12

We appreciate all reviewers and the area chair for their valuable feedback and constructive comments. We are encouraged by the reviewers’ recognition of our work’s key strengths:

- Focus on efficiency of video processing, a key bottleneck of video LLMs. (QvJS)
- The method is well-designed (JwZp), solves the problem well (vfkR), and is easy to follow (GXf1).
- Highly impressive results (JwZp) demonstrate a strong tradeoff on performance vs efficiency (JwZp, QvJS, vfkR) and SOTA performance (GXf1). The work presents fair and credible comparisons (JwZp), is evaluated on diverse benchmarks (JwZp, QvJS), and includes detailed ablation studies (QvJS).
- The paper is well-organized (vfkR, GXf1). Figures are intuitive to understand the benefits of the proposed method (QvJS, vfkR).

In summary, our proposed FastVID is a novel, training-free approach for accelerating Video LLMs. FastVID adaptively identifies highly redundant segments and preserves representative and distinctive tokens, enabling efficient yet accurate pruning. FastVID can be seamlessly applied to diverse architectures, including LLaVA-OneVision (Table 1 in the main paper), LLaVA-Video (Table 2 in the main paper), Qwen2-VL (Table 7 in the appendix), and Qwen2.5-VL (responses to JwZp's Q1 and vfkR's Q1). Across various short- and long-video benchmarks, FastVID achieves SOTA performance. Under extreme compression rates, FastVID significantly outperforms existing methods. This provides a practical and efficient solution for fast Video LLMs.

We sincerely thank all reviewers for their positive assessment of our work, noting that **all preliminary scores were on the positive side**. We are further encouraged that **all reviewers confirmed that their concerns have been fully addressed**. We will incorporate the new discussions and experiments in the revised version.

Finally, we extend our gratitude to all reviewers and the area chair for their time and effort in evaluating our work.

---

### Decision · Program_Chairs · 2025-09-17

**Decision:**

Accept (poster)

**Comment:**

This paper introduces a dynamic token-pruning approach that exploits spatiotemporal redundancy to reduce computing cost for video LLMs. Initial reviews were on the positive side, and after rebuttal all reviewers recommended acceptance. In spite of minor concerns about some technical and experimental details, the reviewers highlighted the technical innovation, strong empirical results, and high quality presentation. The AC concurs with the reviewers' post-rebuttal assessment and recommend acceptance. The AC encourages the authors to incorporate the rebuttal materials into the camera-ready version.